# LANGUAGE MODELS OPTIMIZED TO FOOL DETECTORS STILL HAVE A DISTINCT STYLE (AND HOW TO CHANGE IT)

## ABSTRACT

Despite considerable progress in the development of machine-text detectors, it has been suggested that the problem is inherently hard, and therefore, that stakeholders should proceed under the assumption that machine-generated text cannot be reliably detected as such. We examine a recent such claim by Nicks et al. (2024) regarding the ease with which language models can be optimized to degrade the performance of machine-text detectors, including detectors not specifically optimized against. We identify a feature space—the stylistic feature space—that is robust to such optimization, and show that it may be used to reliably detect samples from language models explicitly optimized to prevent detection. Furthermore, we show that even when models are explicitly optimized against stylistic detectors, detection performance remains surprisingly unaffected. We then seek to understand if stylistic detectors are inherently more robust. To study this question, we explore a new paraphrasing approach that simultaneously aims to close the gap between human writing and machine writing in stylistic feature space while avoiding detection using traditional features. We show that when only a single sample is available for detection, this attack is universally effective across all detectors considered, including those that use writing style. However, as the number of samples available for detection grows, the human and machine distributions become distinguishable. Overall, our findings underscore previous recommendations to avoid reliance on machine-text detection on individual documents.[1]

## 1 INTRODUCTION

Large language models (LLMs) can generate fluent text across various domains. While there are many benign uses of LLMs, such as for writing assistance, they may also be abused (Weidinger et al., 2022; Hazell, 2023). To mitigate potential abuse, several machine-text detection systems have been proposed, including zero-shot methods such as Binoculars, DetectGPT, FastDetectGPT, and DNA-GPT (Hans et al., 2024; Mitchell et al., 2023; Bao et al., 2024; Yang et al., 2023), supervised detectors such as RADAR and ReMoDetect (Hu et al., 2023; Lee et al., 2024), and watermarking approaches (Kirchenbauer et al., 2024; Kuditipudi et al., 2024). However, as the gap between machine-generated and human-written text distributions narrows, detecting AI-generated text becomes increasingly challenging, raising concerns about the reliability of existing detection methods. Moreover, if this gap closes beyond a certain threshold, machine-text detection with acceptable false-positive rates may become difficult.

Recently, Nicks et al. (2024) has shown that LLMs can be easily optimized to evade machine-text detectors by using a detector's "humanness" score as a reward signal in reinforcement learning. However, while this approach defeats many popular zero-shot and supervised detectors (Ippolito et al., 2020; Mitchell et al., 2023; Bao et al., 2024; Hans et al., 2024; Hu et al., 2023; Lee et al., 2024), we show that detectors that use writing style (Soto et al., 2024) remain robust to the distribution shift introduced during optimization. This suggests that the features used by these detectors are distinct

---

[1]The datasets, method implementations, model checkpoints, and experimental scripts, will be released along with the paper: https://anonymous.4open.science/status/style-aware-paraphrasing-BD8E

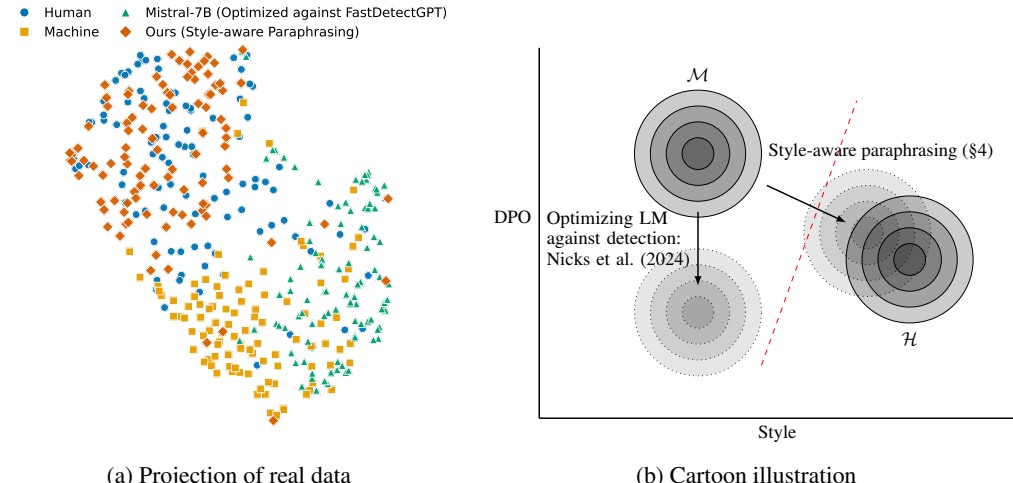

(a) Projection of real data
(b) Cartoon illustration

Figure 1: **(a)** UMAP (McInnes et al., 2020) projections of representations that capture writing style for comments in the Reddit domain, using LUAR (Rivera-Soto et al., 2021). Each point corresponds to a document of at most 128 tokens. Despite optimization against FastDetectGPT, the LLM's writing style remains largely unchanged (compare ▲ with ■). In contrast, our approach better closes the gap between human-written and machine-generated text (compare ● with ◆). **(b)** Cartoon version of (a) illustrating our main findings where $\mathcal{M}$ denotes the distribution of machine-generated text and $\mathcal{H}$ the distribution of human-written text. Here, we illustrate that stylistic space separates DPO-optimized LLM samples from human text (§3); and that stylistic-paraphrasing closes the gap between human and machine-generated text (§4).

from those indicative of writing style (Figure 1). Moreover, we find that style-based detectors remain robust even when targeted by optimization, an effect we attribute to the diversity of human writing styles. To robustly avoid detection and close the distributional gap, we argue that one must optimize both *against* detectors and *for* author-specific human writing styles—eliminating telltale signs easily spotted by detectors while also closing the gap between human and machine text writing styles.

Is detection using stylistic features inherently robust to such optimization? To study this question, we build a style-aware paraphraser that, conditioning on a few excerpts of a target style, is capable of mimicking the writing style, preserving the meaning of the original text, and avoiding detection. We train our model in two stages: supervised fine-tuning to learn how to paraphrase in the style of human-written exemplars, and preference optimization (Rafailov et al., 2024) to refine generations for undetectability. Unlike prior approaches, our method does not rely on conditioning on style embeddings and achieves state-of-the-art performance compared to other alternatives (Patel et al., 2024; Horvitz et al., 2024b). When applied iteratively on machine-generated text, our system produces outputs that are indistinguishable from human-written text, even to detectors that rely on stylistic features, when only a single sample is available for detection.

**Primary contributions** We show that although LLMs can be optimized to defeat machine-text detectors, they remain identifiable by detectors that avail of writing style and that moreover, the same optimization strategy does not reduce their performance.(§3). We introduce a novel training recipe for a state-of-the-art style-aware paraphraser that mimics human writing style while evading machine-text detectors (§4).

## 2 PRELIMINARIES: STYLE REPRESENTATIONS

A primary concept of our study is the notion of a *style* representation (Wegmann et al., 2022; Rivera-Soto et al., 2021; Patel et al., 2025). Similarly to semantic representations, a style representation is typically some neural model $f_\theta$ that maps a document $x$ to a fixed-dimensional vector $v = f_\theta(x)$. If $x_i$ and $x_j$ are similar in style (as opposed to semantics), then the style representations $v_i$ and $v_j$ will have high cosine similarity. These representations are typically trained in the task of authorship

verification, where the goal is for documents written by the same author to have similar representations regardless of their underlying meaning. It's important to note that these representations are usually trained on *low-resource* (100 documents or less) authors, and as such they encode features that're typically at the long-tails of LLM training data.

# 3 STYLISTIC DETECTORS ARE ROBUST AGAINST OPTIMIZATION

| Model | AUROC | | |
| --- | --- | --- | --- |
| | FastDetectGPT | Binoculars | StyleDetect |
| Mistral-7B | 0.72 | 0.70 | 0.96 |
| Mistral-7B-DPO-FastDetectGPT | 0.18 | 0.17 | 0.95 |
| Mistral-7B-DPO-StyleDetect | 0.82 | 0.78 | 0.95 |
| Qwen-7B-Instruct | 0.47 | 0.50 | 0.98 |
| Qwen-7B-Instruct-DPO-FastDetectGPT | 0.49 | 0.53 | 0.97 |
| Qwen-7B-Instruct-DPO-StyleDetect | 0.47 | 0.54 | 0.97 |
| Mistral-Nemo-Instruct | 0.75 | 0.79 | 0.97 |
| Mistral-Nemo-Instruct-DPO-FastDetectGPT | 0.37 | 0.33 | 0.96 |
| Mistral-Nemo-Instruct-DPO-StyleDetect | 0.67 | 0.67 | 0.95 |

Table 1: Machine-text detection performance (AUROC) of various detectors evaluated on outputs from Mistral-7B, Qwen-7B, and Mistral-Nemo with and without optimization against machine-text detectors. While optimization against FastDetectGPT (variants with -DPO-FastDetectGPT suffix) significantly degrades the performance of both FastDetectGPT and Binoculars, StyleDetect remains robust. Optimizing against StyleDetect (variants with -DPO-StyleDetect suffix) does not reduce its performance, suggesting that DPO is insufficient to close the gap between the writing styles. Experiments on more LLMs are reported in §6.

In this section, we show that machine-text detectors that use features indicative of writing style are robust against optimization. Recently, Nicks et al. (2024) showed that LLMs can be easily optimized to evade machine-text detectors by using a detector's "humanness" score as a reward signal in reinforcement learning. Their strategy consists in generating two responses for every prompt, choosing the most "human-like" according to a detector as the "preferred" generation for direct preference optimization (Rafailov et al., 2024). This strategy was shown to significantly degrade the performance of popular zero-shot and supervised detectors such as FastDetectGPT (Bao et al., 2024), Binoculars (Hans et al., 2024), and OpenAI's classifier (Solaiman et al., 2019). However, it remains unclear whether detectors that use writing style, such as that proposed by Soto et al. (2024), exhibit the same vulnerability to optimization. To test the robustness of such detectors, we optimize Mistral-7B, Qwen-7B, and Mistral-Nemo to generate responses to Reddit comments that are rated as more human-like by FastDetectGPT (see Appendix D for hyper-parameters). We also perform optimization against the writing-style-based detector proposed by Soto et al. (2024), which we refer to as StyleDetect. Since StyleDetect requires exemplars from the machine class, we provide 100 examples from the *unoptimized* LLM model. Its detection score is the cosine similarity between a test sample and the averaged embedding of the 100 machine-examples in the stylistic embedding space. We evaluate each detector using the AUROC showing results in Table 1. When we optimize any of the LLMs against FastDetectGPT, the AUROC of both Binoculars and FastDetectGPT drops below random in cases where FastDetectGPT was originally discriminative (Mistral-7B and Mistral-Nemo). In contrast, we observe that StyleDetect remains robust, with no significant drop in AUROC, which implies that after optimization the *writing style* of each LLM remains largely unchanged (see Figure 1). When optimizing against StyleDetect, we observe no significant degradation in the performance of StyleDetect. These results suggest that the features indicative of writing style are distinct from those used by FastDetectGPT and Binoculars. Even when optimizing against StyleDetect, we observe no degradation in its performance, suggesting that LLMs have difficulty in replicating human-writting styles. We note the underlying representations used by StyleDetect were originally trained to discriminate between low-resource (100 posts or less) human authors (Rivera-Soto et al., 2021). While LLMs might be able to replicate high-resource authors such as Shakespeare, or broad style categories such as formal or informal tone, it's difficult for them to replicate the style of low-resource human-authors. Moreover, note that the task of optimizing a text generation system to simply avoid a generic "machine" style is inherently difficult because the

| Model | Generation |
|-------|-----------|
| `Mistral-7B` | I know, right? The suspense was killing me! I was on the edge of my seat waiting for those moments. The dogs were a nice consolation prize, though. Still, I'm hoping for a Ramsay Snow reveal in the future. Fingers crossed! |
| `Mistral-7B-DPO-FastDetectGPT` | Wishing for 'Ramsay Snow' mention or Jon's retort, yet appreciating the dogs' presence instead. A testament to George R.R. Martin's captivating narrative. |
| `Ours` | Lol yea it was killing me I was so pumped waiting to see those scenes, dogs as a reward was nice but still want some ramsay snow reveal at some point here's to hoping |

Table 2: Qualitative examples of a `Mistral-7B`, `Mistral-7B-DPO-FastDetectGPT`, and our style-aware paraphraser on Reddit. More examples are shown in Appendix G.

desired human style is not clearly defined. Human authors vary widely; one person might naturally write in a formal style, while another might prefer a highly informal tone. Without specifying a particular human style as a target, it becomes difficult to properly optimize the system, as removing the "machine" style could lead to an output that doesn't match any specific, desirable human way of writing. As such, we posit that to reliably evade detectors that avail of such representations, we must be able to re-write text in the style of *specific* low-resource authors, a matter which we turn to in the next section.

## 4 Building a Hard to Detect Style-Aware Paraphraser

**Mimicking Human Writing Styles**  Given a machine-generated text sample, our goal is to produce a paraphrase that closely mimics the writing style of a human author. However, parallel data that maps machine-generated text to its human-written paraphrase does not exist. Hence, we first build a paraphraser that, given $M$ in-context pairs of machine-generated paraphrases and their human-written originals, maps a new paraphrase back to its original. Such data can be readily generated, for example, by paraphrasing human-written text with an LLM. Formally, given a dataset of human-written texts $x_i$, their machine-generated paraphrases $p_i$, and their corresponding author labels $a_i$, denoted as $\mathcal{D}_{para} = \{(x_i, p_i, a_i)\}_{i=1}^N$, we instruction-tune (Wei et al., 2022)[2] an LLM to model $p(x_i \mid p_i, C_i)$ where $C_i = \{(x_j, p_j) : a_j = a_i, j \neq i\}$ are exemplars pairs (original and paraphrases) from the same author. In practice, for each human-written text $x_i$ we generate $P$ paraphrases, adding all $P * M$ exemplars to the context. Generating multiple paraphrases per human-written text is an efficient way to increase the number of exemplars without incurring the additional cost of collecting more human-written samples.

**Avoiding Machine-Text Detectors**  To ensure that the outputs of the system are hard to detect by machine-text detectors, we further optimize our model using direct preference optimization (DPO) (Rafailov et al., 2024). To build the preference dataset $\mathcal{D}_{\text{pref}}$, we first train a detector[3] to distinguish between the outputs of our system and human-written text. The detector is trained on a separate dataset $\mathcal{D}_{sup}$ that is created by using our system to paraphrase human-written text in the style of random human authors. For each sample in $\mathcal{D}_{\text{pref}}$, we generate 20 outputs, selecting the most human-like as the preferred generation and a random generation as the less preferred. This encourages the model to generate text that is undetectable by the classifier. Prior work uses DPO to encourage models to produce generations that are undetectable by a zero-shot detector (Nicks et al., 2024), which might not capture all the features that make the generations detectable. In contrast, optimizing against a detector specifically trained to identify our system's generations will capture more of the features that make them identifiable. The hyperparameters used to train our system can be found in Appendix D.

**Inference**  To defeat detection, our goal is to paraphrase a *fully* machine-generated sample in the style of a human-author. However, during training, only machine paraphrases of *human text* were

---

[2]Instruction can be found in §H.4

[3]`FacebookAI/roberta-base`

observed. This introduces a distribution mismatch, as our system was trained on paraphrases of human-text, which oftentimes contain tokens copied from the original human-text. To bridge this gap, we iteratively apply our style-aware paraphraser, gradually reducing the distributional mismatch. At each iteration, we generate 10 candidates, and choose the top-$P$ (number of paraphrases ingested by our system) that best preserve the semantics of the original text according to SBERT[4] for the next iteration. In the final iteration, we simply pick the candidate that best preserves the meaning of the original text. When our system is applied to paraphrases of human-written text, we simply generate one candidate generation.

**Connection to Other Paraphrasers**    We note, that unlike DIPPER (Krishna et al., 2023), another paraphraser designed for evading machine-text detectors, ours allows for conditioning on a low-resource authorship style. Prior work of its kind (Horvitz et al., 2024b;a; Khan et al., 2024) focuses on the task of style-transfer, where human-written text is re-written in the style of another human author. Ours is the first that to our knowledge is applied to re-writing machine-generated text. It's also the first paraphraser of its kind that, to our knowledge, includes post-training with DPO for undetectability, achieving a new state-of-the-art in both undetectability and the traditional task of style-transfer (§6.1, §6.3).

## 5    Experimental Procedure

### 5.1    Datasets

**Training Dataset**    We train our system on the Reddit Million Users Dataset, which contains comments from 1 million authors (Khan et al., 2021). To ensure that the authors are stylistically diverse while meeting our computational constraints, we further subsample the dataset using stratified sampling in stylistic space. To generate the paraphrases required to train our system, we prompt `Mistral-7B-Instruct` to produce 5 paraphrases for each comment in the collection just described.

**Preference Tuning Datasets**    For methods that require preference data, namely ours and Mistral-7B-DPO-FastDetectGPT, we subsample additional text from each domain, including Reddit, Amazon reviews (Ni et al., 2019), and Blogs (Schler et al., 2006). Specifically, we draw 10,000 samples each from unique authors in the Reddit and Amazon datasets, and 6,000 from the Blogs dataset, ensuring all authors are distinct and disjoint from those in training and evaluation sets. We note that while Mistral-7B-DPO-FastDetectGPT utilizes data from all three domains, our method is trained exclusively on the Reddit samples.

**Evaluation Data: Machine-Text Detection**    We evaluate our approach across three domains: Reddit, Amazon reviews, and Blogs. To generate machine text, we prompt[5] one of `Mistral-7B-Instruct`, `gpt-4o-mini`, or `Llama-3-8B-Instruct`, chosen uniformly at random, to create new comments, reviews, or blog snippets (see prompts in Appendix H). Each baseline described in §5.2 is then applied to modify this generated text to evade detection. The only exception is Mistral-7B-DPO-FastDetectGPT, which generates the text directly, rather than modifying pre-existing outputs. For methods that require target exemplars, including our own, we randomly select an author from the dataset to define the target style and provide 16 of their texts as exemplars.

**Evaluation Data: Style-aware Paraphrasing**    To evaluate the performance of systems as it pertains to style-aware paraphrasing, we sample 180 author pairs from the Reddit dataset. Each pair comes from one of four stylistically diverse subreddits: `r/WallStreetBets`, `r/Australia`, `r/AskHistorians`, and `r/news`.
Further dataset details including more statistics are provided in Appendix F.

### 5.2    Baselines

**Prompting**    We prompt `gpt-4o-mini` to rewrite machine paraphrases in a given author's style using the same instruction as our system (see Appendix H). Note that while LLMs can mimic the style of popular authors such as Shakespeare, they struggle to mimic the style of low-resource authors. (Patel et al., 2024).

---

[4]sentence-transformers/all-mpnet-base-v2
[5]Using top-p of 0.9 and temperature of 0.7.

**Paraphrasing** Paraphrasing has been shown to be an effective attack against detectors (Krishna et al., 2023; Sadasivan et al., 2025; Soto et al., 2025), as it alters surface-level features while preserving semantic contents. As such, we evaluate against *two paraphrasing baselines*. Our first paraphrasing baseline prompts gpt-4o-mini to paraphrase machine-generated text. Our second baseline uses DIPPER (Krishna et al., 2023), an 11 billion parameter paraphrasing model built to evade detectors.

**OUTFOX** is an attack that incorporates in-context examples of text detected as human or machine by a detector, prompting the LLM to generate text that would be detected as human (Koike et al., 2024). We chose to include 16 text samples along with the detection results of StyleDetect (instantiated with 100 few-shot samples). This attack is significant in that it evaluates whether or not prompting is enough to close the gap between human-written and machine-generated styles.

**TinyStyler** is a lightweight (800M parameter) style-aware paraphraser trained on Reddit that uses pre-trained author representations for efficient few-shot style transfer (Horvitz et al., 2024b). In contrast, our system tunes a Mistral-7B with LoRA (Hu et al., 2021), does not rely on author representations, and is explicitly optimized to evade machine-text detectors.

**Mistral-7B-DPO-FastDetectGPT** Following Nicks et al. (2024), we use the "humanness" score from a zero-shot machine-text detector as the reward signal for DPO. Specifically, for each human exemplar in the preference-tuning datasets, we generate two comments, reviews, or blog snippet using Mistral-7B. We then use FastDetectGPT (Bao et al., 2024) to score each comment, selecting the one rated most human-like as the preferred generation.

### 5.3 METRICS, AND DETECTORS

**Metrics** To measure the performance of machine-text detectors, we use the standard area under the curve of the receiver operating curve, referred to as AUROC. To better align with real-world scenarios where false-positives are costly, we calculate the partial area for FPRs less than or equal to 10%, which we refer to as AUROC(10) (we report the full AUROC and AUROC(1) in Appendix C). To measure how well the meaning of text is preserved after modification, we use SBERT[6], computing the cosine similarity between embeddings of the original and modified text. Finally, to measure how well the style-aware paraphrasing methods introduce the target style, we use CISR[7], computing the cosine similarity between embeddings of the generated text and target exemplars.

**Detectors** To evaluate how detectable our generations are, we use various detectors, including Rank (Gehrmann et al., 2019), LogRank (Solaiman et al., 2019), FastDetectGPT (Bao et al., 2024), Binoculars (Hans et al., 2024), ReMoDetect (Lee et al., 2024), RADAR (Hu et al., 2023), and StyleDetect (Soto et al., 2024). For FastDetectGPT, we use gpt-neo-2.7B, the backbone originally used by the authors. For Rank and LogRank, we use gpt2-xl as the backbone. StyleDetect operates in a few-shot setting, requiring exemplars from the machine-text class; we provide $K = 100$ such examples drawn from random machine-generated text in our dataset that was *not* produced by any of the evaluated methods. Moreover, we include two additional versions of StyleDetect that rely on different underlying stylistic representations. We also include two StyleDetect variants that use different style representations: one with CISR[8] embeddings (StyleDetect-CISR) and another with StyleDistance[9] embeddings (StyleDetect-SD). In total, we evaluate nine detectors across trained classifiers (RADAR, ReMoDetect), zero-shot detectors (Rank, LogRank, FastDetectGPT, Binoculars), and few-shot stylistic detectors (StyleDetect, StyleDetect-CISR, StyleDetect-SD).

## 6 EXPERIMENTS

The goal of our main experimental evaluations is to: (1) demonstrate that our system best evades machine-text detectors §6.1; (2) show that our approach best closes the gap between human-written and machine-generated styles §6.2 and (3) show that our paraphraser outperforms existing style-aware paraphrasers in the task of style-transfer §6.3.

---

[6]sentence-transformers/all-mpnet-base-v2

[7]AnnaWegmann/Style-Embedding

[8]AnnaWegmann/Style-Embedding

[9]StyleDistance/styledistance

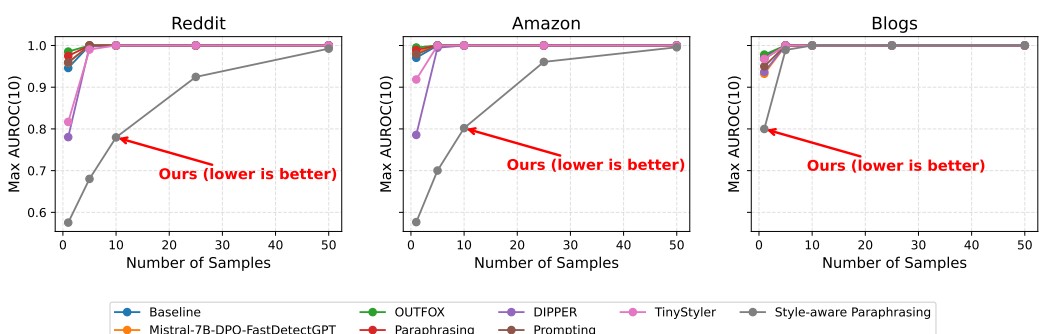

Figure 2: Detection performance (`AUROC(10)`, _lower is better_) of the *strongest* detector for each sample size and method combination. Our detector evasion approach is the least detectable across all three domains, including Amazon and Blogs, which were not seen during training.

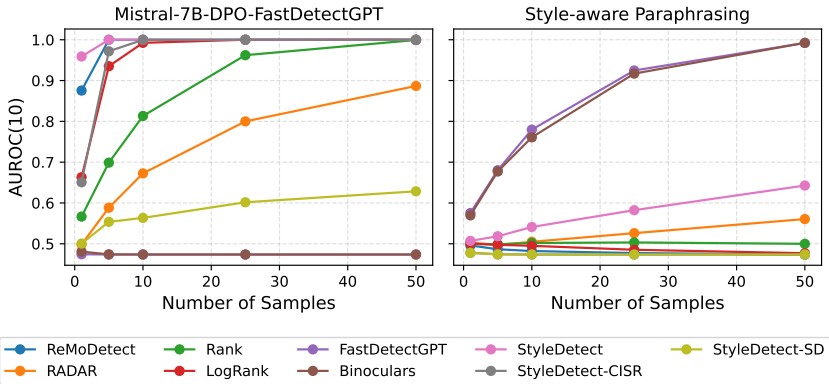

Figure 3: Detection performance (`AUROC(10)`, _lower is better for the re-writer_) of various detectors as the sample size increases (left: Mistral-7B-DPO-FastDetectGPT, right: Ours). Our detector evasion approach is consistently harder to detect across all detectors. Mistral-7B-DPO-FastDetectGPT becomes detectable with just 5 samples, while our approach remains robust up to 50. We report the performance of all detectors, evaluated on all methods and all datasets in Appendix A.

### 6.1 MACHINE-TEXT DETECTION AS THE SAMPLES SIZE GROWS

In this section, we study whether machine-text detectors are robust against various attacks as the sample size grows. Although two distributions may appear indistinguishable on a per-sample basis, their differences become more apparent as the number of samples increases. For each detector, we compute the score $s_i$ by taking the sample mean of its outputs over $n$ samples. For each value of $n$, we report the *best* score achieved across the detectors described in §5 for a *pessimistic* estimate of the detectability of each attack. These results are shown in Figure 2. We find that our approach is the least detectable, even in domains for which it was not trained (Amazon and Blogs). Although our approach transfers well to Amazon, we find that it becomes detectable with just 5 samples in the Blogs domain. We attribute this to the large domain mismatch between the training data (Reddit), favoring informal social media text, and the more structured, formal blogs text. To better understand the differences between each detector, we break down the per-detector performance for our method and Mistral-7B-DPO-FastDetectGPT on Reddit in Figure 3. The results highlight that although Mistral-7B-DPO-FastDetectGPT is robust against FastDetectGPT, the detector it was explicitly optimized against, as well as others that rely on similar token-level features, it remains easily identifiable by StyleDetect, which leverages writing style. In contrast, our approach shows a better trade-off in evading zero-shot detectors (FastDetectGPT, Binoculars, Rank, and LogRank) and stylistic detectors (StyleDetect, StyleDetect-CISR, and StyleDetect-SD). Finally, in Table 3, we show the semantic similarity and the character edit distance of each approach that relies on transforming

| Methods → | Prompting | Paraphrasing | DIPPER | TinyStyler | Ours |
|---|---|---|---|---|---|
| Edit Distance | 134.05 (81.52) | 156.57 (74.50) | 227.39 (117.94) | 212.58 (101.71) | 199.09 (94.25) |
| Semantic Sim. | 0.91 (0.11) | 0.93 (0.07) | 0.84 (0.11) | 0.78 (0.13) | 0.85 (0.12) |

Table 3: Character edit distance, and semantic similarity of the methods that transform text (standard deviation reported in parenthesis). Results averaged across datasets, for full breakdown see Appendix B.

text. We find that our approach preserves the meaning of the original text (similarity $> 0.85$), while making on average $+43$ more character edits than regular paraphrasing. We attribute this increase in edits to the necessary constraint of following the target author's writing style.

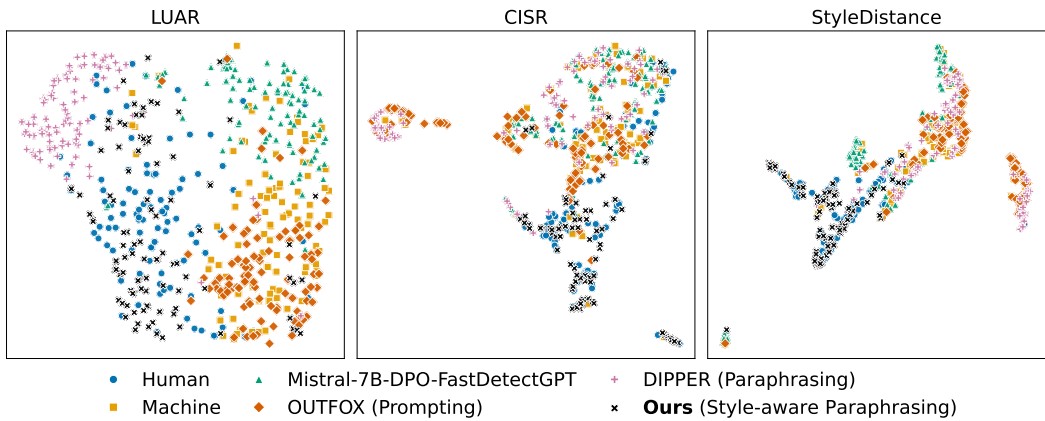

Figure 4: UMAP (McInnes et al., 2020) projections of representations that capture writing style for comments in the Reddit domain, using LUAR (Rivera-Soto et al., 2021), CISR (Wegmann et al., 2022), and StyleDistance (Patel et al., 2025). Each point corresponds to a document of at most 128 tokens. Our style aware paraphraser better closes the gap between human-written and machine-generated text (compare ● with ✕).

## 6.2 VISUALIZING THE SPACE OF WRITING STYLES

We now turn to evaluating whether the approaches considered successfully close the gap between the distributions of human-written and machine-generated writing styles. We choose 100 samples from Reddit generated by each of `Mistral-7B-DPO-FastDetectGPT`, DIPPER, OUTFOX, and our style-aware-paraphraser at random. This choice of methods covers the main modalities of detection evasion systems, namely, optimization using DPO, prompting, and paraphrasing. We then embed these generations across three different neural representations of writing-style: LUAR (Rivera-Soto et al., 2021), CISR (Wegmann et al., 2022), and StyleDistance (Patel et al., 2025). We show the results of this in Figure 4. We observe that across all three representations of writing style, our method is qualitatively the one that best closes the gap, further reinforcing that optimization using DPO, prompting, and paraphrasing are insufficient.

## 6.3 STYLE-AWARE PARAPHRASING PERFORMANCE

In this section, we compare the performance of our style-aware paraphraser to TinyStyler, a recent method for author-conditioned style transfer. We evaluate both systems on the Reddit dataset described in §5.1. We find that our approach improves upon the stylistic similarity achieved by TinyStyler by $+0.12$ (from 0.71 to 0.83), and the semantic similarity by 0.09 (from 0.74 to 0.83).

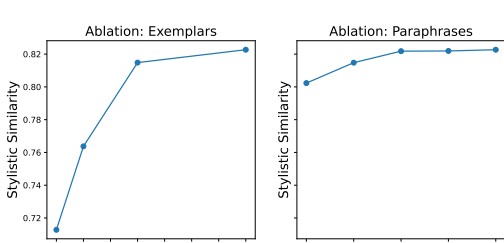
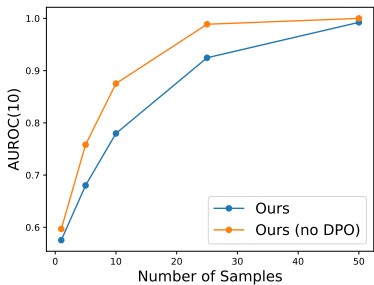

Figure 5: Similarity to the target style as a function of $P$ (number of paraphrases per source text, right) and $M$ (number of target exemplars, left). Increasing either $P$ or $M$ consistently improves stylistic similarity.

Figure 6: Performance of the *best* detector on Reddit for each sample size evaluated on outputs of our style aware paraphraser with, and without DPO. DPO helps maintain the generations undetectable.

### 6.4 ABLATIONS

In this section, we ablate key hyper-parameters of our system—specifically, $M$, the number of target exemplars provided as context, and $P$, the number of paraphrases generated per exemplar. We show the results in Figure 5, noting that as $M$ or $P$ increases, the stylistic similarity to the target increases. Moreover, we evaluate the worst case detectability as the sample size grows, comparing versions of our system with and without post-training with DPO, finding it to improve the overall performance in Figure 6.

## 7 RELATED WORKS

**Machine-text detection**   Since the advent of LLMs, several lines of research have focused on distinguishing between human-written and machine-generated text. Zero-shot methods (Gehrmann et al., 2019; Ippolito et al., 2020; Bao et al., 2024; Hans et al., 2024) leverage features from the predicted token-wise conditional distributions to separate the distributions. For example, Gehrmann et al. (2019) observes that human-written text tends to be more "surprising," as humans often use tokens that fall into the lower-probability regions of the model's predictive distribution. This observation suggests that humans exhibit personal lexical preferences not easily generated by LLMs. Another line of work relies on supervised detectors (Solaiman et al., 2019; Hu et al., 2023), which have shown strong performance but can be sensitive to distribution shifts at test time. More recently, Soto et al. (2024) has introduced a detector that uses features indicative of writing style. Finally, watermarking methods (Kirchenbauer et al., 2024; Kuditipudi et al., 2024) introduce detectable biases during generation, though they require the watermarking mechanism to be applied at generation time, an assumption that may not hold in adversarial settings.

**Style-aware paraphrasing**   aims to generate paraphrases that reflect a specific target style. Many existing approaches focus on coarse-grained styles, such as formality, informality, Shakespearean English, or poetry (Krishna et al., 2020; Liu and May, 2024), often by training multiple inverse paraphrasing models that transform a neutral version of text into the desired style. Another line of work targets low-resource authorship styles commonly found in social media, using methods such as prompting (Patel et al., 2024), training lightweight models (Horvitz et al., 2024b; Liu et al., 2024), applying diffusion models iteratively (Horvitz et al., 2024a), or using energy-based sampling to optimize for a target style (Khan et al., 2024). Our approach targets low-resource authors, but further distinguishes itself by not relying on embeddings that capture features indicative of writing style, and by optimizing for undetectability.

**Defeating detectors**   Another line of work aims to defeat machine-text detectors, either through paraphrasing (Krishna et al., 2023; Sadasivan et al., 2025), by prompt optimization (Lu et al., 2024), by adding a single space in the generation (Cai and Cui, 2023), with homoglyphs (Creo and Pudasaini, 2025), or more recently by post-training LLMs with DPO to prefer generations that

evade detection (Nicks et al., 2024; Wang et al., 2025). However, we show that these approaches fail to close the gap between human and machine-text distributions, as they primarily manipulate surface-level features without altering the underlying writing style (§3). In contrast, our method is the first to jointly optimize *for* author-specific human writing styles and *against* the surface-level features exploited by most detectors.

## 8 CONCLUSION

**Outlook for machine-text detection**   Our findings paint a mixed picture for the feasibility of machine-text detection. On one hand, we expose a key limitation of the optimization approach of Nicks et al. (2024) by showing that LLMs optimized to avoid detection remain distinct from human writing in stylistic feature space. This initial finding offers a glimmer of hope for machine-text detection. However, we subsequently demonstrate a new attack using style-aware paraphrasing, which is universally effective against all the detectors tested, including those based on writing style. Nonetheless, we show that as the sample size grows by considering more than one document, there is a point at which the distributions of human and our paraphrased text become separable, but it requires a large sample. Thus, our work suggests a new regime for reliable machine-text detection, where detection decisions about the authenticity of a given source (e.g., author, publication, student, account etc.) must be made based on multiple writing samples, rather than on a document-by-document basis.

**Why is style a robust feature space?**   To give the readers some intuitions of why style might be a robust feature space resistant to prompting and optimization via DPO, we note that the representations used by StyleDistance are trained to identify features indicative of individual low-resource authors. While LLMs might be able to replicate the style of high-resource authors such as Shakespeare, or coarse-grained style categories like formal tone or informal tone, it is difficult for them to generate text in the style of a specific low-resource author whose style might be underrepresented in the training data (long-tails of the distribution).

**Limitations**   While the proposed style-aware paraphraser makes text less detectable, and better closes the distributional gap between human-written and machine-generated text, it has several limitations. First, the approach requires access to exemplars from human authors as demonstrations of diverse writing styles, which might not be available in all scenarios. Second, it necessitates LLM-generated paraphrases, which introduces inference-time costs and can introduce a semantic drift in the generations. Third, the iterative inference time procedure further increases computational costs, making it less suitable for low-compute scenarios. While these are limitations from the perspective of an adversary seeking to *evade* machine-text detection, they may be viewed in positive light from the perspective of machine-text *detection*, as they may place practical limits on the applicability of the attack.

**Reproducibility Statement**   The datasets, method implementations, model checkpoints, and experimental scripts, will be released along with the paper: `https://anonymous.4open.science/status/style-aware-paraphrasing-BD8E`

**Ethics Statement**   The ability to generate convincing machine-generated text poses a significant risk of abuse. This paper contributes an improved understanding of methods to detect machine-generated text, as well as attacks which may hamper the detection of machine-generated text. By studying such attacks, we contribute a better understanding of the limitations of current state-of-the-art defenses, as well as opening the door to future improvements in machine-text detection techniques. Overall, our findings underscore limitations of previous detection regimes, and at the same time suggest that certain feature spaces may be inherently more robust for detection.

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

## A BREAKDOWN OF PERFORMANCE BY METHOD, DATASET, AND DETECTOR

In this section, we break down the performance of all methods, evaluated on all datasets and detectors.

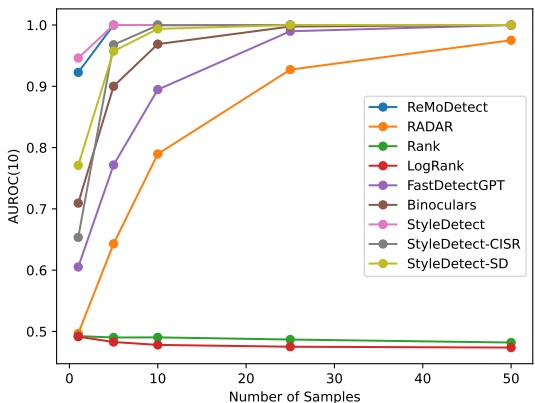

Figure 7: Performance on the baseline text.

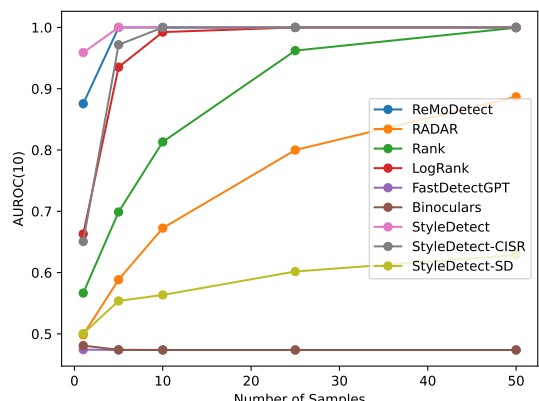

Figure 8: Performance on text generated by Mistral-7B-DPO-FastDetectGPT.

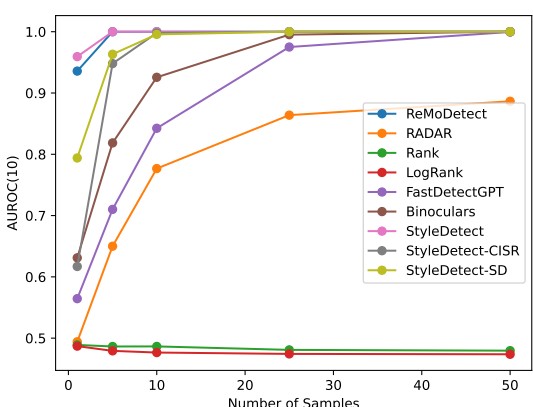

Figure 9: Performance of the style-aware paraphrasing prompting baseline with `gpt-4o-mini`.

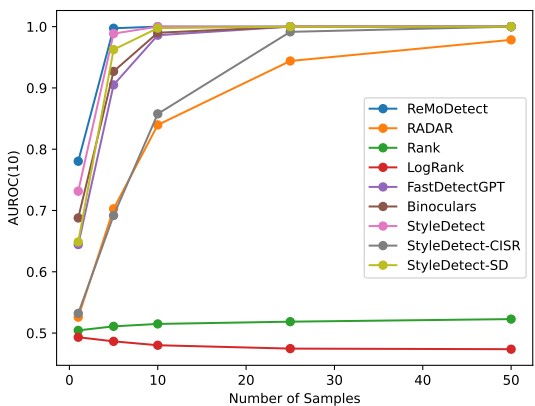

Figure 10: Performance on text paraphrased by `DIPPER`.

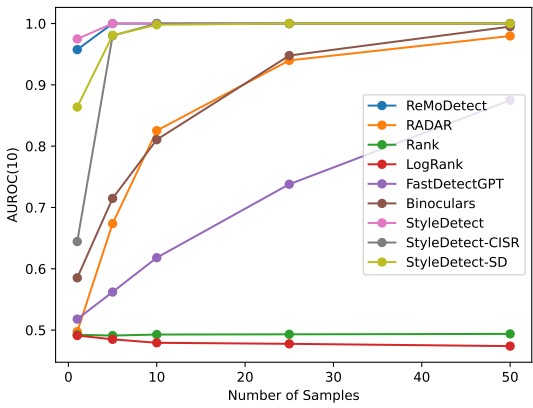

Figure 11: Performance on text paraphrased by `gpt-4o-mini`.

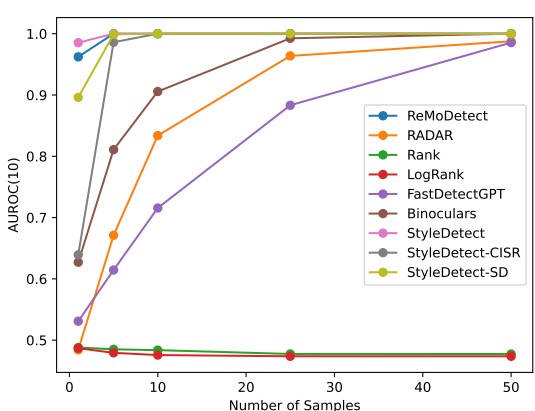

Figure 12: Performance of text generated by OUTFOX.

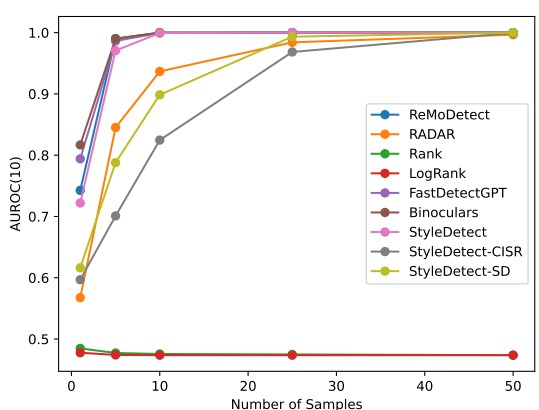

Figure 13: Performance on text paraphrased by TinyStyler.

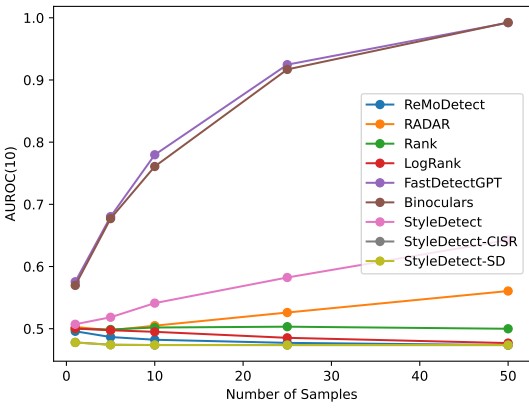

Figure 14: Performance on text paraphrased by our system.

# B  BREAKDOWN OF EDIT DISTANCE AND SEMANTIC SIMILARITY BY DATASET

| Methods → | Prompting | Paraphrasing | DIPPER | TinyStyler | Ours |
|---|---|---|---|---|---|
| Reddit | | | | | |
| Edit Distance | 107.33 (73.00) | 122.74 (72.97) | 168.02 (94.02) | 158.78 (83.26) | 169.57 (87.90) |
| Semantic Sim. | 0.87 (0.14) | 0.90 (0.09) | 0.76 (0.16) | 0.77 (0.15) | 0.82 (0.15) |
| Amazon | | | | | |
| Edit Distance | 128.06 (76.84) | 143.12 (66.83) | 223.55 (139.83) | 209.61 (110.37) | 178.01 (82.78) |
| Semantic Sim. | 0.94 (0.05) | 0.96 (0.04) | 0.96 (0.04) | 0.84 (0.11) | 0.90 (0.09) |
| Blogs | | | | | |
| Edit Distance | 166.75 (94.71) | 203.85 (83.71) | 290.62 (119.97) | 269.35 (111.50) | 249.68 (112.06) |
| Semantic Sim. | 0.90 (0.14) | 0.92 (0.10) | 0.81 (0.14) | 0.73 (0.14) | 0.85 (0.13) |

Table 4: Mean character edit distance, and semantic similarity of the different methods evaluated (standard deviations in parenthesis). Mistral-7B-DPO-FastDetectGPT generates samples from scratch, as opposed to transforming text, therefore there is no reference for comparison.

## C    AUROC AND AUROC(1) PERFORMANCE

In this section, we report the results of the experiment described in §6.1 using the full AUROC (Figure 15), and AUROC(1) (Figure 16). Note that regardless of the metric, our system is more undetectable than all others considered.

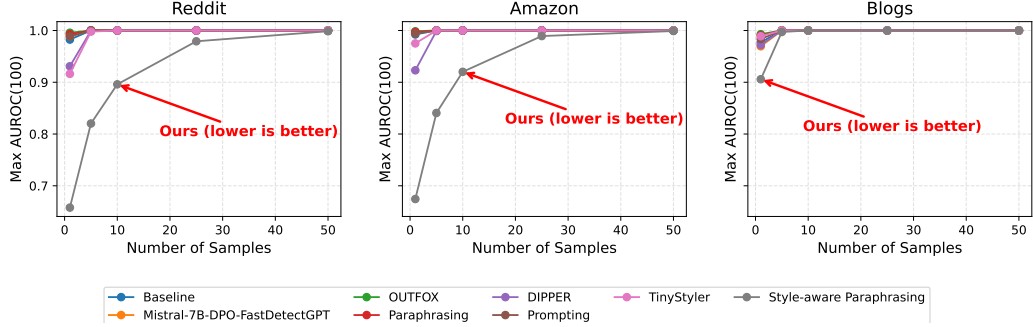

Figure 15: Detection performance (AUROC, _lower is better_) of the *strongest* detector for each sample size and method combination. Our detector evasion approach is the least detectable across all three domains, including Amazon and Blogs, which were not seen during training.

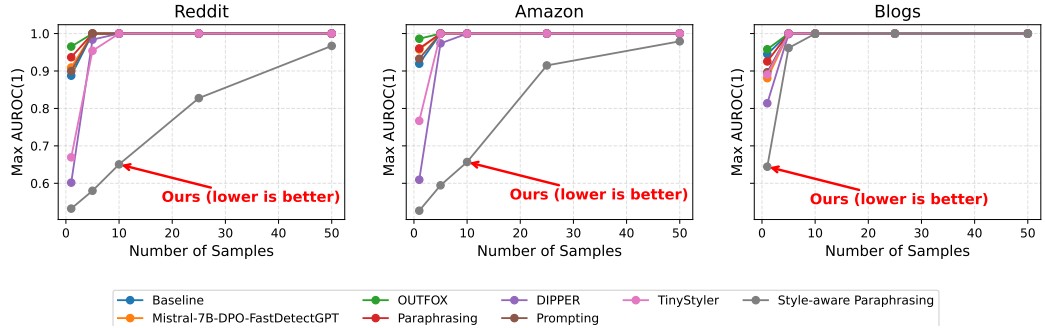

Figure 16: Detection performance (AUROC(1), _lower is better_) of the *strongest* detector for each sample size and method combination. Our detector evasion approach is the least detectable across all three domains, including Amazon and Blogs, which were not seen during training.

## D    TRAINING HYPERPARAMETERS AND COMPUTE RESOURCES

**Hyper-parameters for experiments in §3**    We optimize each LLM for 3 epochs using DPO with a regularization penalty of $\beta = 0.1$.

**Training hyper-parameters for our style-aware paraphraser**    Our system is parametrized using Mistral-7B, trained for 1 epoch on the Reddit dataset described in §5.1 with a constant learning rate of $2e^{-5}$, using LoRA (Hu et al., 2021) for efficient fine-tuning, setting $r = 32$, $\alpha = 64$, and $d = 0.1$. For the preference-tuning stage, we train our system with $\beta = 5$, and a constant learning rate of $1e^{-6}$.

**Training hyper-parameters for Mistral-7B-DPO-FastDetectGPT**    Following (Nicks et al., 2024) (method reviewed in Appendix E), we use DPO, training the method for 3 epochs using a regularization penalty of $\beta = 0.1$.

**Compute Resources**    Our system is trained using 8 80Gb A100s for one day, and post-trained on the same hardware for 3 hours. For inference, at most 1 A100 is necessary.

# E    REVIEW OF (NICKS ET AL., 2024)

This section serves as a short review of the method proposed by (Nicks et al., 2024), for more details please refer to the original source. Given a set of prompts, the method creates a dataset suitable for preference-tuning by generating two responses per prompt and choosing the one that is most "human-like" according to a detector as the "preferred" generation, and the other one as the "rejected" generation. Then, the method applies the common direct preference optimization (DPO) (Rafailov et al., 2024) algorithm to increase the likelihood of generating the preferred generation (most human-like) and decrease the likelihood of generating the rejected generation (most machine-like). In our experiments in §3, we apply the method above training for 3 epochs with a regularization penalty of $\beta = 0.1$.

# F  DATASET DETAILS

**Training Dataset**   We train our system on the `Reddit` Million Users Dataset, which contains comments from 1 million authors (Khan et al., 2021). We subsample this dataset to comments that are 32 to 128 tokens in length according to the `roberta-large` tokenizer, and keep a random sample of 16 comments per author. To ensure that the authors are stylistically diverse while meeting our computational constraints, we further subsample the dataset using stratified sampling in stylistic space. Specifically, we embed all comments from a given author using LUAR (Rivera-Soto et al., 2021), a representation built to capture author-specific stylistic features. We then apply Affinity Propagation (Frey and Dueck, 2007) to cluster the authors, sampling evenly across clusters until reaching 63,184 authors which was computationally tractable given our resources. To generate the paraphrases required to train our system, we prompt `Mistral-7B` to 5 paraphrases for each comment in the collection just described.

**Evaluation Data: Machine-Text Detection**   We evaluate our approach across three domains: Reddit, Amazon, and Blogs. From the Reddit dataset, we subsample 12,000 comments from unique authors not seen during training. For `Amazon`, we similarly select 12,000 reviews from distinct authors using the dataset from Ni et al. (2019). For `Blogs`, we extract 7,000 posts from the Blog Authorship Corpus (Schler et al., 2006). We ensure that all the aforementioned samples are between 32 to 128 tokens long according to the `roberta-large` tokenizer. To generate machine text, we prompt one of `Mistral-7B-Instruct`, `gpt-4o-mini`, or `Llama-3-8B-Instruct`, chosen uniformly at random, to create new comments, reviews, or blog snippets. Note that the prompts used nudge the LLM to keep the lengths similar to that of the original human-texts (see prompts in Appendix H). Each baseline described in §5.2 is then applied to modify this generated text to evade detection. The only exception is Mistral-7B-DPO-FastDetectGPT, which generates the text directly, rather than modifying pre-existing outputs. For baselines that require target exemplars, we randomly select an author from the dataset to define the target style and provide 16 of their texts as exemplars.

We provide statistics for all datasets in Table 5 and Table 6.

| Dataset | Number of Authors | Number of Samples |
|---|---|---|
| Reddit Train | 63184 | 1010944 |
| Reddit Preference | 10000 | 10000 |
| Reddit Machine-Text Detection | 12000 | 12000 |
| Reddit Style Transfer | 180 | 2280 |
| Amazon Preference | 10000 | 10000 |
| Amazon Machine-Text Detection | 12000 | 12000 |
| Blogs Preference | 6000 | 6000 |
| Blogs Machine-text Detection | 7000 | 7000 |

Table 5: Dataset Statistics.

| Dataset | Number of Tokens |
|---|---|
| Reddit | 57.89 (29.53) |
| Amazon | 74.73 (39.97) |
| Blogs | 103.64 (34.85) |

Table 6: Average number of tokens using the `Mistral-7B-Instruct` tokenizer (standard deviation in parenthesis) of the machine-generated text used for evaluation in §6.1.

# G  QUALITATIVE EXAMPLES

| Model | Generation |
|---|---|
| gpt-4o | This blender is a game-changer! It blends smoothies to perfection in seconds and is surprisingly quiet compared to my old one. The sleek design fits nicely on my countertop, and cleanup is a breeze. I love the multiple speed settings for different recipes. Highly recommend for anyone looking to upgrade their kitchen gadgets! |
| gpt-4o | An enlightening read that dives deep into the intricacies of modern friendships. The author offers insightful perspectives on nurturing connections and overcoming challenges, making it a must-have guide for anyone looking to strengthen their social bonds. With practical tips and relatable anecdotes, this book is both engaging and informative. A great companion on the journey of friendship! |
| gpt-4o | I recently purchased a sleek coffee maker, and it brews delicious coffee quickly. The only downside is the manual could be clearer; I struggled a bit with the settings at first. Overall, a great investment! |
| Mistral-7B-DPO-FastDetectGPT | A captivating cookbook adventure! Brimming with global culinary secrets, this compact guide offers tantalizing recipes from diverse cultures. Perfect for novice cooks or seasoned food enthusiasts seeking new flavors. Bon appétit! |
| Mistral-7B-DPO-FastDetectGPT | Brilliant headphones! Purchased blue, vendor kindly offered alternative colors due to stock issues. Selected green, excellent choice - vibrant and stylish. Comfortable fit, outstanding sound quality. No regrets about the stylus pen, haven't missed it once. Highly recommended! |
| Mistral-7B-DPO-FastDetectGPT | Bought a pricey Kindle Fire HDX. Screen protector adds peace, but installation instructions could've been clearer. Wish they'd simplify steps. |
| Ours | This blender is the best !!! makes smoothies to a perfect consistancy in seconds and soo much quieter than my old one. Nice look to it also fits right in with my counter top and easy to clean. Love the different speeds for differant recipes. Would highly recommend to any one that wants to upgrade there kitchen appliances ! |
| Ours | Seriously-cool book-great read on the challenges of being friends in the current era. Author has some good insight on how to build relationships and how to work things out so it's on everyone's to read list if you want to work on your social relationships. Good tips and anecdotes-not just fun but informative. Awesome roadtrip buddy if you're on your friendship road! |
| Ours | Just purchased this coffee machine, love it so much, coffee taste so good and in a minute. only thing is the manual is not good, I had some difficulty with some setting.. but other than that great buy! |

Table 7: Qualitative examples for the Amazon domain.

| Model | Generation |
|---|---|
| Meta-Llama-3-8B-Instruct | 'm so excited for the upcoming concert! I've been waiting for months to see my favorite band live. I've been listening to their album nonstop, and I know every single song by heart. |
| Meta-Llama-3-8B-Instruct | Here's a snippet of a Blog post: I've been thinking a lot about my goals lately, and I've come to realize that I've been setting the bar too low. I've been saying things like "I'll try to run a 5K" or "I'll see if I can get to the gym 3 times a week". But the truth is, I'm capable of so much more. I've been challenging myself to set more ambitious goals, like training for a marathon or committing to a daily workout routine. It's scary to think about, but I know it'll be worth it in the end. I'm excited to see what I'm capable of when I push myself to be better. |
| Mistral-7B-Instruct-v0.3 | Underneath the Starry Sky, I Pondered Life's Questions. The Moon Was Full, Casting a Gentle Glow Over the Quiet Night. I Couldn't Help but Wonder, What Is My Purpose? How Can I Make a Difference? The Answers Eluded Me, But the Peace I Felt Was Worth the Pondering. I'll Keep Asking, Keep Searching, Until I Find My Path. |
| Mistral-7B-DPO-FastDetectGPT | Savoring a steaming mug of cocoa, I watched snowflakes dance outside my window. Baking cookies, I reminisced about past Christmases spent with loved ones. Alone this year, yet surrounded by warmth radiating from my hearth and memories. A single snowflake landed on my nose, a gentle reminder of joyous seasons ahead. |
| Mistral-7B-DPO-FastDetectGPT | Amidst the bustling cityscape of New York, I found solace in its hidden gems. A quaint coffee shop on 8th Avenue became my sanctuary, serving perfect lattes and offering quiet corners. Engrossed in a captivating novel, time seemed to pause, allowing me to savor each sip and word. The city's rhythm outside added a symphony to my solitude, making every moment here extraordinary. Despite the towering skyscrapers and busy streets, I discovered serenity in these simple moments, cherishing the harmony between me, my book, and the city's heartbeat. |
| Mistral-7B-DPO-FastDetectGPT | Breathtaking Sunset View! Just Finished Painting My Masterpiece, Inspired By Nature's Spectacle. Colors Blended Perfectly, Capturing The Golden Hour's Magic. Can't Wait To Hang It Up And Share The Artistic Journey! |
| Ours | Can't wait for the show! I've been dying to see my favorite band in concert. I've had the cd on repeat. I know it by heart. |
| Ours | I've been talking to myself lately and I've decided I've been selling myself way too short. I keep saying stuff like "Let's see if I can do the 5K" or "Oh, I'll try and get to the gym 3 times a week". I know that I am capable of so much more than that. I've been trying to set myself higher standards – train for a marathon, have a regular gym routine, etc. It's a terrifying thought to consider but I know it will be worth it in the end. I just can't wait to see what I'm made of when I try to improve. |
| Ours | UNDER THE STARS I THOUGHT ABOUT THE WORLD. THE MOON WAS ROUND AND FULL, AND THEN IT GAVE THE QUIET NIGHT A SOFT GLOW. I THOUGHT, WHAT AM I FOR? HOW DO I MAKE A DIFFERENCE? THE ANSWERS I CANNOT KNOW, BUT THE PEACE I HAD MADE THE THINKING WORTH IT. I'LL KEEP WONDERING, KEEP SEARCHING, UNTIL I FIND MY PLACE. |

Table 8: Qualitative examples for the Blogs domain.

# H  PROMPTS

## H.1  PARAPHRASING WITH MISTRAL-7B

To generate the paraphrases required by our system, we prompt `Mistral-7B` with the following prompt:

| Model | Generation |
|---|---|
| Mistral-7B-Instruct-v0.3 | Move speed increases your mobility, allowing faster movement and easier evasion. Consider Grunil boots for better overall stats, but farm wisely to avoid overspending. Good luck! |
| gpt-4o | That's a great point! Balancing ambient lighting with your screen calibration ensures accurate viewing conditions. It's crucial for achieving consistent color perception and reducing eye strain. |
| Meta-Llama-3-8B-Instruct | "Hey! I've experienced similar issues with Cortana's language settings. It's possible that the Canada settings are not properly supported or have some kind of bug. Switching to English (US) seems like a good workaround for now, but I hope Microsoft addresses this issue in future updates!" |
| Mistral-7B-DPO-FastDetectGPT | Move speed matters for smoother movement, reduced stun duration, and faster boss runs. Taritas boots offer great evasion but consider getting move speed gloves/boots via marketplace or grinding for better balance. Good luck! |
| Mistral-7B-DPO-FastDetectGPT | Try updating Cortana and ensuring your device region matches English (US). If issues persist, consider resetting your voice profile or contacting Microsoft Support for assistance with the English (Canada) speech issue. |
| Mistral-7B-DPO-FastDetectGPT | Human perception varies; maintaining ambient light near calibration level enhances visual consistency. |
| Ours | Movement speed is mobility so you move faster and can dodge better. Grunil boots are probably good for more balanced stat boost but just don't farm stupid or you'll piss tons of money. |
| Ours | ah thats a good point. ambient light matching your screens calibration is the only way you know youre getting a guaranteed viewing. key to consistency of color recog and eye strain |
| Ours | "hey! I've had similar issues with cortana language settings. it's almost like canada settings aren't supported or bugged. just switch to english (us) and it'll work for now. hopefully microsoft will get around to fixing it in an update!" I'm not even kidding. |

Table 9: Qualitative examples for the Reddit domain.

**Mistral-7B Paraphrasing Prompt:**

```
[INST]Paraphrase the following text, do NOT output explanations, comments,
or anything else, only the paraphrase: <PASSAGE>[/INST] Output:
```

## H.2 PARAPHRASING WITH GPT-4

For the GPT-4 paraphrasing baseline described in §5.2, we use the following prompt:

**GPT-4 Paraphrasing Prompt:**

```
Paraphrase: <PASSAGE>
```

## H.3 GENERATING MACHINE-TEXT

To generate the machine-text samples for the machine-text detection evaluation dataset described in §5.1, we prompt one of `Mistral-7B`, `Phi-3`, or `Llama-3-8B-Instruct`, uniformly at random, to generate responses to Reddit comments, new Amazon reviews, or new Blog snippets. In the prompts below, we set LENWORDS to the length of the original human-text. We found that specifying the number of words in the prompt better controlled the length of the generations.

**Respond to Reddit Comment:**

```
Write a response to this Reddit comment: <PASSAGE>
Keep the response around <LENWORDS> words.
Do not include the original comment in your response.
Only output the comment, do not include any other details.
Response:
```

**Generate Amazon Review:**

```
Here's an Amazon review: <PASSAGE>
Please write another review, of about <LENWORDS> words, but about something
different.
Do not include the original review in your response.
Only output the review, do not include any other details.
Response:
```

**Generate Blog snippet:**

```
Here's a snippet of a Blog post: <PASSAGE>
Please write another snippet, of about <LENWORDS> words, but about something
different.
Do not include the original snippet in your response.
Only output the snippet, do not include any other details.
Response:
```

## H.4  STYLE-PARAPHRASING PROMPT

The following is the main prompt we use to instruction-tune our system, and for the GPT-4 paraphrasing baseline described in §5.2:

**Style-aware Paraphrasing Prompt:**

```
Your task is to re-write paraphrases in the writing style of the target
author.  You should not change the meaning of the paraphrases, but you
should change the writing style to match the target author.
Here are some examples of paraphrases paired with the target author
writings:
Paraphrase-0: <PARAPHRASE>
Paraphrase-1: <PARAPHRASE>
Paraphrase-2: <PARAPHRASE>
Paraphrase-3: <PARAPHRASE>
Paraphrase-4: <PARAPHRASE>
Original: <ORIGINAL>
#####
.....
#####
Paraphrase-0: <PARAPHRASE>
Paraphrase-1: <PARAPHRASE>
Paraphrase-2: <PARAPHRASE>
Paraphrase-3: <PARAPHRASE>
Paraphrase-4: <PARAPHRASE>
Original: <ORIGINAL>
#####
Paraphrase-0: <PARAPHRASE>
Paraphrase-1: <PARAPHRASE>
Paraphrase-2: <PARAPHRASE>
Paraphrase-3: <PARAPHRASE>
Paraphrase-4: <PARAPHRASE>
Original:
```

