# OpenReview forum: "Language Models Optimized to Fool Detectors Still Have a Distinct Style (And How to Change It)"
_ICLR.cc/2026/Conference — Submitted to ICLR 2026_

### Official Review · Reviewer_3Y7h · 2025-10-30

**Soundness:** 2
**Presentation:** 2
**Contribution:** 2
**Rating:** 2
**Confidence:** 3

**Summary:**

This paper identifies a stylistic feature space that can be used to detect LLM-generated text, in a way that remains robust against LLMs specifically optimized to evade detection using existing optimization techniques. It then proposes a training recipe using SFT and DPO to fine-tune LLMs to both mimic the writing style of human-written text and avoid detection, and shows that text generated by such fine-tuned LLMs evades detectors that rely on stylistic features, but only when a limited number of generated samples are presented to the detectors.

**Strengths:**

- The authors study an interesting problem of machine-text detection with important implications for preventing misuse of LLMs. The idea of fine-tuning LLMs to both mimic human writing styles and avoid detection conceptually makes sense.
- The authors evaluate various detectors and baseline attacks, showing comprehensive results.

**Weaknesses:**

- Overall, I think the paper lack clarity. Please see Questions.
- The evaluation setup in Section 2 is not clear. How is the optimization against each detector done? What parameters did you use when you optimized the LLMs following Nicks et al.? Are the parameters the same across all LLMs evaluated?
- It’s good to have the AUROC(10) metric, but you should also report the usual AUROC score. Additionally, if the argument is that false-positives are costly in real-world scenarios, FPR of 10% is still quite high, what would the AUROC be for lower FPRs? (e.g., 1% FPR used in StyleDetect?)
- Concerns about overclaiming: according to Figure 3, your approach is more easily detectable against FastDetectGPT and Binoculars (even when only a single sample is available for detection), which contradicts claims "In contrast, our approach is more universally undetectable across all detectors tested" and "Our detector evasion approach is consistently harder to detect across all detectors." The fact that with 10 samples, FastDetectGPT and Binoculars reach nearly 0.8 AUROC(10) on text generated by your approach limits its applicability to a very few number of samples, it’d be more ideal if there isn’t such a “trade-off” between detectability by FastDetectGPT/Binoculars and StyleDetect.

Minor presentation issues include:
- Duplicate sentences in lines 267 and 270
- Typo in table 1 caption "variatns with -DPO-StyleDetect suffix"
- Grammar error in line 288 "can best evades", missing "(2)" in line 289
- In Table 2 and Appendix E, it'd be nice to include the original human-written text to compare with the LLM-generated samples and show that your approach does better

**Questions:**

- Could you clarify the training and evaluation setup? Specifically, section 4.1 says that samples are drawn from Reddit, Amazon reviews, and Blogs datasets to construct the preference tuning dataset, but in line 300 you mention that your approach is not trained in "Amazon and Reddit" domains
- Could you clarify what type of detector was trained for distinguishing between the outputs of your system and human-written text, before using this detector to construct the preference dataset (referred to in "Avoiding Machine-Text Detectors" part of Section 3)? How big is D_sup? How did you measure the level of "human-like" in generated outputs?
- In section 4.1, when you generate 5 paraphrases for each sampled human-written text to construct the training dataset, do you evaluate the semantic similarity between the paraphrases and the original? What about the diversity of the paraphrases?
- In section 4.2, what's the difference between the prompting baseline and the first paraphrasing baseline, since both prompt gpt-4o-mini to paraphrase machine-generated text?
- In section 4.3, you describe the inference process of your approach, since it is specific to your approach, it’s better to describe it in the method section instead. Additionally, how the inference process works is not entirely clear to me, what system prompt do you use for your style-aware paraphraser at each iteration? How many iterations do you do and how do you decide how many iterations to do?

---

> ### Author Response · Authors · 2025-11-21
>
> We thank the reviewers for their thoughtful comments and constructive feedback. Note that we’ve updated our manuscript, the section numbers referred to in the rebuttal are those of the revised manuscript. To facilitate the discussion, we briefly summarize our three primary contributions. First, we provide a comprehensive analysis demonstrating that current DPO-optimized attacks (e.g., Nicks et al.) fail to remove stylistic "fingerprints," rendering them vulnerable to style-based detection. Second, we demonstrate that even when DPO is used to optimize specifically against style-based detectors, the outputs still remain identifiable by said detectors. Third, we introduce a novel Style-Aware Paraphraser that bridges the gap between style transfer and evasion by simultaneously optimizing for specific authorship styles and undetectability, achieving state-of-the-art performance when compared against other attacks (Figure 2, Section 6.1) and in the traditional task of style-transfer (Section 6.3).
>
>
> > The evaluation setup in Section 2 is not clear. How is the optimization against each detector done? What parameters did you use when you optimized the LLMs following Nicks et al.? Are the parameters the same across all LLMs evaluated?
>
> For optimization against each detector, we follow the method of Nicks et. al. where for each prompt we generate two samples from the model, picking the most “human-like” as evaluated by the detector as the preferred generation and the less human-like as the rejected generation. We use a \beta of 0.1 (as stated in Appendix D), and train for three epochs. The parameters are kept fixed across all LLMs evaluated. We modified the manuscript to make these details clearer.

---

> > ### Author Response · Authors · 2025-11-21
> >
> > > It’s good to have the AUROC(10) metric, but you should also report the usual AUROC score. Additionally, if the argument is that false-positives are costly in real-world scenarios, FPR of 10% is still quite high, what would the AUROC be for lower FPRs? (e.g., 1% FPR used in StyleDetect?)
> >
> > We have included the full AUROC and AUROC(1) tables in the updated Appendix. As shown below, at $N=1$ (the standard deployment scenario for many detectors), our method achieves significantly lower AUROC(1) (0.53 on Reddit) compared to baselines (0.67-0.97). While detection performance naturally rises with N (supporting that multi-sample detection is necessary), our method remains the hardest to detect across all datasets.
> > # AUC(1)
> >
> >
> > ### Reddit
> >
> > |                              |    1 |    5 |   10 |   25 |   50 |
> > |:-----------------------------|-----:|-----:|-----:|-----:|-----:|
> > | Baseline                     | 0.89 | 1.00 | 1.00 | 1.00 | 1.00 |
> > | Mistral-7B-DPO-FastDetectGPT | 0.91 | 1.00 | 1.00 | 1.00 | 1.00 |
> > | OUTFOX                       | 0.97 | 1.00 | 1.00 | 1.00 | 1.00 |
> > | Paraphrasing                 | 0.94 | 1.00 | 1.00 | 1.00 | 1.00 |
> > | DIPPER                       | 0.60 | 0.98 | 1.00 | 1.00 | 1.00 |
> > | Prompting                    | 0.90 | 1.00 | 1.00 | 1.00 | 1.00 |
> > | TinyStyler                   | 0.67 | 0.95 | 1.00 | 1.00 | 1.00 |
> > | Ours     | 0.53 | 0.58 | 0.65 | 0.83 | 0.97 |
> >
> > ### Amazon
> >
> > |                              |    1 |    5 |   10 |   25 |   50 |
> > |:-----------------------------|-----:|-----:|-----:|-----:|-----:|
> > | Baseline                     | 0.92 | 1.00 | 1.00 | 1.00 | 1.00 |
> > | Mistral-7B-DPO-FastDetectGPT | 0.96 | 1.00 | 1.00 | 1.00 | 1.00 |
> > | OUTFOX                       | 0.99 | 1.00 | 1.00 | 1.00 | 1.00 |
> > | Paraphrasing                 | 0.96 | 1.00 | 1.00 | 1.00 | 1.00 |
> > | DIPPER                       | 0.61 | 0.97 | 1.00 | 1.00 | 1.00 |
> > | Prompting                    | 0.93 | 1.00 | 1.00 | 1.00 | 1.00 |
> > | TinyStyler                   | 0.77 | 1.00 | 1.00 | 1.00 | 1.00 |
> > | Ours     | 0.53 | 0.59 | 0.66 | 0.91 | 0.98 |
> >
> > ### Blogs
> >
> > |                              |    1 |    5 |   10 |   25 |   50 |
> > |:-----------------------------|-----:|-----:|-----:|-----:|-----:|
> > | Baseline                     | 0.95 | 1.00 | 1.00 | 1.00 | 1.00 |
> > | Mistral-7B-DPO-FastDetectGPT | 0.88 | 1.00 | 1.00 | 1.00 | 1.00 |
> > | OUTFOX                       | 0.96 | 1.00 | 1.00 | 1.00 | 1.00 |
> > | Paraphrasing                 | 0.93 | 1.00 | 1.00 | 1.00 | 1.00 |
> > | DIPPER                       | 0.81 | 1.00 | 1.00 | 1.00 | 1.00 |
> > | Prompting                    | 0.90 | 1.00 | 1.00 | 1.00 | 1.00 |
> > | TinyStyler                   | 0.89 | 1.00 | 1.00 | 1.00 | 1.00 |
> > | Ours     | 0.64 | 0.96 | 1.00 | 1.00 | 1.00 |
> >
> > # AUC(full)
> >
> > ### Reddit
> >
> > |                              |    1 |    5 |   10 |   25 |   50 |
> > |:-----------------------------|-----:|-----:|-----:|-----:|-----:|
> > | Baseline                     | 0.98 | 1.00 | 1.00 | 1.00 | 1.00 |
> > | Mistral-7B-DPO-FastDetectGPT | 0.99 | 1.00 | 1.00 | 1.00 | 1.00 |
> > | OUTFOX                       | 1.00 | 1.00 | 1.00 | 1.00 | 1.00 |
> > | Paraphrasing                 | 0.99 | 1.00 | 1.00 | 1.00 | 1.00 |
> > | DIPPER                       | 0.93 | 1.00 | 1.00 | 1.00 | 1.00 |
> > | Prompting                    | 0.99 | 1.00 | 1.00 | 1.00 | 1.00 |
> > | TinyStyler                   | 0.92 | 1.00 | 1.00 | 1.00 | 1.00 |
> > | Ours     | 0.66 | 0.82 | 0.90 | 0.98 | 1.00 |
> >
> > ### Amazon
> >
> > |                              |    1 |    5 |   10 |   25 |   50 |
> > |:-----------------------------|-----:|-----:|-----:|-----:|-----:|
> > | Baseline                     | 0.99 | 1.00 | 1.00 | 1.00 | 1.00 |
> > | Mistral-7B-DPO-FastDetectGPT | 1.00 | 1.00 | 1.00 | 1.00 | 1.00 |
> > | OUTFOX                       | 1.00 | 1.00 | 1.00 | 1.00 | 1.00 |
> > | Paraphrasing                 | 1.00 | 1.00 | 1.00 | 1.00 | 1.00 |
> > | DIPPER                       | 0.92 | 1.00 | 1.00 | 1.00 | 1.00 |
> > | Prompting                    | 0.99 | 1.00 | 1.00 | 1.00 | 1.00 |
> > | TinyStyler                   | 0.98 | 1.00 | 1.00 | 1.00 | 1.00 |
> > | Ours     | 0.67 | 0.84 | 0.92 | 0.99 | 1.00 |
> >
> > ### Blogs
> >
> > |                              |    1 |    5 |   10 |   25 |   50 |
> > |:-----------------------------|-----:|-----:|-----:|-----:|-----:|
> > | Baseline                     | 0.98 | 1.00 | 1.00 | 1.00 | 1.00 |
> > | Mistral-7B-DPO-FastDetectGPT | 0.97 | 1.00 | 1.00 | 1.00 | 1.00 |
> > | OUTFOX                       | 0.99 | 1.00 | 1.00 | 1.00 | 1.00 |
> > | Paraphrasing                 | 0.99 | 1.00 | 1.00 | 1.00 | 1.00 |
> > | DIPPER                       | 0.97 | 1.00 | 1.00 | 1.00 | 1.00 |
> > | Prompting                    | 0.98 | 1.00 | 1.00 | 1.00 | 1.00 |
> > | TinyStyler                   | 0.99 | 1.00 | 1.00 | 1.00 | 1.00 |
> > | Ours     | 0.91 | 1.00 | 1.00 | 1.00 | 1.00 |

---

> ### Author Response · Authors · 2025-11-21
>
> > Concerns about overclaiming: according to Figure 3, your approach is more easily detectable against FastDetectGPT and Binoculars (even when only a single sample is available for detection), which contradicts claims "In contrast, our approach is more universally undetectable across all detectors tested" and "Our detector evasion approach is consistently harder to detect across all detectors." The fact that with 10 samples, FastDetectGPT and Binoculars reach nearly 0.8 AUROC(10) on text generated by your approach limits its applicability to a very few number of samples, it’d be more ideal if there isn’t such a “trade-off” between detectability by FastDetectGPT/Binoculars and StyleDetect.
>
> While we agree with the reviewer to tone down 'universal' language, we wish to clarify the trade-off. Existing optimization methods (Nicks et al.) drastically fail against stylistic detectors (Table 1). Our method is the first to evade both zero-shot and stylistic detectors effectively in the single-sample regime (Figure 2, N=1). While Figure 3 shows our method is detectable with sufficient samples ($N \ge 10$), this aligns with our finding that reliable detection requires multiple samples.
>
> > Could you clarify the training and evaluation setup? Specifically, section 4.1 says that samples are drawn from Reddit, Amazon reviews, and Blogs datasets to construct the preference tuning dataset, but in line 300 you mention that your approach is not trained in "Amazon and Reddit" domains
>
> We apologize for the confusion caused by an error in Section 5.1. Our method was only trained/preference-tuned on Reddit. The text 'namely ours and Mistral-7B-DPO...' should have stated that the baseline (Mistral-7B-DPO) utilized all domains (following Nicks et al.), while 'Ours' utilized only Reddit. We will correct this in the final manuscript. This confirms that Amazon and Blogs are indeed unseen domains for our method, validating the transferability claims."
>
> > Could you clarify what type of detector was trained for distinguishing between the outputs of your system and human-written text, before using this detector to construct the preference dataset (referred to in "Avoiding Machine-Text Detectors" part of Section 3)? How big is D_sup? How did you measure the level of "human-like" in generated outputs?
>
> We trained a FacebookAI/RoBERTa-base model which has 125 million parameters. D_sup contains 12,000 human-written texts and 12,000 machine-generated texts. “Human-like” is measured using the outputs of the detectors following Nicks et. al., picking the least detectable one as the more “human-like” one. We’ve modified the manuscript to clarify these details.
>
> > In section 4.1, when you generate 5 paraphrases for each sampled human-written text to construct the training dataset, do you evaluate the semantic similarity between the paraphrases and the original? What about the diversity of the paraphrases?
>
> We do not evaluate the semantic similarity or diversity of the training data. Note that our style-aware paraphrasing task requires mapping input paraphrases (which are neutral in style) to a diverse set of output paraphrases (which exhibit various human styles). For this task, a consistent (low-diversity) set of neutral-styled inputs is preferred because mapping from a limited set of alternatives to a diverse set is inherently easier than mapping between two diverse sets. Moreover, we found the semantic similarity of the final output to be competitive with that of the other approaches considered:
> | **Methods →**     | Prompting        | Paraphrasing     | DIPPER           | TinyStyler       | Ours            |
> |-------------------|------------------|------------------|------------------|------------------|-----------------|
> | **Edit Distance** | 134.05 (81.52)   | 156.57 (74.50)   | 227.39 (117.94)  | 212.58 (101.71)  | 199.09 (94.25)  |
> | **Semantic Sim.** | 0.91 (0.11)      | 0.93 (0.07)      | 0.84 (0.11)      | 0.78 (0.13)      | 0.85 (0.12)
>
>
> > In section 4.2, what's the difference between the prompting baseline and the first paraphrasing baseline, since both prompt gpt-4o-mini to paraphrase machine-generated text?
>
> As stated in Section 5.2, the prompting baseline uses gpt-4o-mini to re-write text in the style of a given author’s style using the prompt in Appendix H.4, while the paraphrasing baseline prompts gpt-4o-mini to simply paraphrase without controlling the style.

---

> > ### Author Response · Authors · 2025-11-21
> >
> > > In section 4.3, you describe the inference process of your approach, since it is specific to your approach, it’s better to describe it in the method section instead. Additionally, how the inference process works is not entirely clear to me, what system prompt do you use for your style-aware paraphraser at each iteration? How many iterations do you do and how do you decide how many iterations to do?
> >
> > We’ve moved the discussion regarding our inference procedure to Section 4 per the reviewers comments. Note that our approach doesn’t require a “system” prompt, it uses the same prompt the model was trained on (Appendix H.4). We do three iterations, as we noted that three was enough to make the generations more human-like in terms of their stylistic similarity to the target author.
> >
> > > Minor presentation issues include:
> >
> > > Duplicate sentences in lines 267 and 270
> >
> > > Typo in table 1 caption "variatns with -DPO-StyleDetect suffix"
> >
> > > Grammar error in line 288 "can best evades", missing "(2)" in line 289
> >
> > > In Table 2 and Appendix E, it'd be nice to include the original human-written text to compare with the LLM-generated samples and show that your approach does better
> >
> > Thank you for your comments, we’ve updated our manuscript to reflect these changes.

---

### Official Review · Reviewer_ZPpT · 2025-10-31

**Soundness:** 3
**Presentation:** 2
**Contribution:** 2
**Rating:** 2
**Confidence:** 3

**Summary:**

The authors show that DPO methods which have been used to circumvent log-likelihood based detectors of machine generated text are ineffective against style-based detectors of machine generated text, even when the DPO method is explicitly designed to evade style-based detectors. They then show that an alternative paraphraser can be effective at evading style-based detectors.

There are essentially three findings:

1) Rewriting a text to evade log-likelihood based detectors of machine generated text does not evade style-based detectors of machine generated text.
2) Attempting to modify the DPO based methods for evading log-likelihood based machine generated text detectors by substituting a style based goal for the log-likelihood based goal is also ineffective at avoiding style based detectors.
3) An alternative (simple but nice) method for evading style based detectors does work.

I think point 1 is good to check, but the result is unsurprising. Point 2 was interesting to me and unintuitive, I would have liked to see more discussion of why this holds (a reason is given on page 1, but I'd have liked to see more justification). The strength of point 3, for me, really depends on how novel the method for evading style based detectors is, I think the authors should do a better job of explaining this.

I think there is quite some chance I will change my review significantly during the discussion stage.

**Strengths:**

The proposed humaniser (as described in section 3.1) is an interesting, simple idea.

The observation that methods to evade log-likelihood based detectors do not evade style based detectors is not surprising, but it is good to see it rigorously shown (one would be very surprised if style based detectors were secretly exploiting artefacts which are visible to log-likelihood based detectors, but it is still good that the authors have ruled this out).

**Weaknesses:**

There are several minor weaknesses which I mention in the questions section below.

Throughout the manuscript the authors use 'detectors' as a proxy for 'log-likelihood based detectors of machine generated text'. There are a lot of clever detectors incorporating different signals than pure log-likelihood (e.g. looking at patterns in log likelihood rather than average log-likelihood), I don't think you need to use all of these different detectors as baselines but I do think you should avoid writing as if average log-likelihood based detectors and style based detectors are the only options.

I would have liked aspects of the experimental design to be more clearly stated (how many tokens were in each text, what parameters were used to generate machine text).

However, I think my main criticism is that I struggled to latch onto which parts of the paper are novel and surprising enough to meet the very high bar of acceptance into ICML. It could be that your humaniser is a simple, elegant new idea which may have applications beyond text detection, my gut feeling is to be impressed by the idea, but if this is the case then I'd like the authors to explain to me how it relates to other paraphrasers (rather than just that the performance is better on one metric), and really make the case for the strength of this idea. It could be the case that the authors have a clever insight about the difference between evading detection methods which aggregate token level scores and evading those that don't, but I would like to see more justification and analysis of the following sentence (How-
ever, while this approach defeats detectors that use token-level features of the predicted conditional
distributions (Ippolito et al., 2020; Mitchell et al., 2023; Bao et al., 2024; Hans et al., 2024), we
show that detectors that use writing style (Soto et al., 2024) remain robust to the distribution shift
introduced during optimization.)

**Questions:**

1) What is special about style that makes it immune to DPO optimisation against style detectors. You suggest on page 1 that the thing about writing style is that it is not present at token level? In that, style detectors are not an aggregate of token level scores. Do you know (or have good reason to think) that this is the reason that DPO optimisation doesn't work? If so, what about other detectors which do not aggregate token level scores, e.g. those based on patterns in log-likelihood (rather than averages of log-likelihood).

2) The proposed humaniser (as described in section 3.1) is an interesting, simple idea. Could you put it in more context as to how it relates to previous paraphrasers. Has a similar idea been tried (successfully or otherwise) elsewhere? Is it generally seen as improving the quality of machine generated text to human eyes? I think this is the strongest idea in the paper, I would like to use the fact that 1) the humanizer is novel and 2) your humaniser could find applications outside of detecting machine generated text as points in your favour, but I'd like you to provide a little more evidence/context for me to do so.

3) After I wrote the previous paragraph I searched for other paraphrase inverters. I found https://aclanthology.org/2025.findings-acl.227/ . Mitigating Paraphrase Attacks on Machine-Text Detection via Paraphrase Inversion
Rafael Alberto Rivera Soto, Barry Y. Chen, Nicholas Andrews. I think this should be cited, and I think it's the same essential idea for evading detection, could you comment.

3) I don't think your claim in the 'main contributions' section that 'optimizing against such detectors does not reduce their performance' is accurate. What you are really saying is that the strategy for optimizing against log-likelihood based detectors does not translate directly to style-based detectors.

4) 'These results suggest that the features indicative of writing style
are distinct from those used by detectors that use features derived from the predicted conditional
distributions.' I think this claim is slightly too strong, I would replace 'features derived from the predicted conditional distributions' with 'log-likelihood'.

5) I would like to know how long the typical texts you are dealing with are (this is useful info when looking at the effectiveness of detectors)

6) What decoding strategies (temp, top-p, top-k etc.) are you using when you generate text from machine. Note,  until recently huggingface enabled top-k=50 by default so unless this has been explicitly controlled for, you may be using top-k generation.

7) You write 'We find that our approach preserves the meaning of
the original text (similarity >0.85)'. Is this a fair conclusion? I have no idea how to interpret it, but note that it is worse than some of the alternative methods.

8) I found it really hard to understand what is meant by 'sample size' in your article. I think the answer is that, when sample size is n, you compute for example the fast detect score of n separate human texts and then compute the mean fast detect score of those n human texts, compared to the mean fast detect score of n machine texts. This is quite an unusual setup, it's not necessarily bad because it's more or less equivalent to just increasing the number of tokens to help detectors, but because we're so primed to imagine you submitting one text for analysis I think you need to go to more effort to stress exactly what you're doing.

9) I think the caption to figure 2 is confusing. When you say detection performance (lower is better), you mean that higher is better for the detector, but lower is better for your paraphraser. I would write something like 'a lower score indicates better performance of the rewriter'.

10) Could you mention text lengths. There is a known problem with the way that Fast-Detect normalises by passage length which can artificially inflate or depress it's performance, see for example TempTest: Local Normalization Distortion and the Detection of Machine-generated Text section 8.2.

---

> ### Author Response · Authors · 2025-11-21
>
> First off, we’d like to thank the reviewer for their thorough review. Before addressing each individual comment and question, we’d like to clarify what we mean by “style” and why we think it’s resistant to optimization, and how our paraphraser is different from other alternatives. Note that we’ve updated our manuscript, the section numbers referred to in the rebuttal are those of the revised manuscript.
>
> Why is style robust to DPO optimization? To give the reviewer some intuitions of why style might be a robust feature space resistant to prompting and optimization via DPO, we note that the representation used by StyleDetect is trained to identify features indicative of individual low-resource (100 posts or less) authors. In fact, StyleDetect’s underlying representations were originally used for authorship attribution[1]. While LLMs might be able to replicate the style of high-resource authors such as Shakespeare, or coarse-grained style categories like formal tone or informal tone, it is difficult for them to generate text in the style of a specific low-resource author whose style might be underrepresented in the training data. Moreover, note that the task of optimizing a text generation system to simply avoid a generic "machine" style is inherently difficult because the desired human style is not clearly defined. Human authors vary widely; one person might naturally write with many unique emojis, while another might not use them at all. Without specifying a particular human style as a target, it becomes difficult to properly optimize the system, as removing the "machine" style could lead to an output that doesn't match any specific, desirable human way of writing. We have expanded the manuscript to include this discussion in Section 3.
>
> Novelty of our style-aware paraphraser:
> Our model introduces style control by re-writing the input text conditioned on text samples from a specific, low-resource human author. This approach fundamentally increases the diversity of generated samples. Previous paraphrasers designed to evade detection (e.g., DIPPER) lack this style control, focusing on masking machine features instead of simulating a specific human distribution.
> Prior work of its kind focuses on the task of style-transfer, where human-written text is re-written in the style of another human author. Ours is the first that to our knowledge is applied to re-writing machine-generated text. It's also the first paraphraser of its kind that, to our knowledge, includes post-training with DPO for undetectability.
> Our paraphraser not only outperforms existing paraphrasers in undetectability but also out-performs the previous state-of-the-art in the task of style-transfer, better adhering to the target style while preserving semantics (see Section 6.3). Moreover, it does so without the need of any pre-trained style representations.
> We’ve included the discussion regarding the novelty of our paraphraser in Section 4.
> We now turn to addressing each of the reviewer’s concerns in a point-by-point manner.
> > Throughout the manuscript the authors use 'detectors' as a proxy for 'log-likelihood based detectors of machine generated text'. There are a lot of clever detectors incorporating different signals than pure log-likelihood (e.g. looking at patterns in log likelihood rather than average log-likelihood), I don't think you need to use all of these different detectors as baselines but I do think you should avoid writing as if average log-likelihood based detectors and style based detectors are the only options.
>
> We agree with the reviewer that there are other detectors that might look at more complicated signals rather than just log-likelihood. We note that we do evaluate against supervised detectors (RADAR and ReMoDetect) that are looking at more than just log-likelihood. However, we agree that there might be other zero-shot detectors that just don’t look at averages of log-likelihood, and therefore we change the language in our revised manuscript to “popular zero-shot and supervised detectors”.

---

> > ### Author Response · Authors · 2025-11-21
> >
> > > What is special about style that makes it immune to DPO optimisation against style detectors. You suggest on page 1 that the thing about writing style is that it is not present at token level? In that, style detectors are not an aggregate of token level scores. Do you know (or have good reason to think) that this is the reason that DPO optimisation doesn't work? If so, what about other detectors which do not aggregate token level scores, e.g. those based on patterns in log-likelihood (rather than averages of log-likelihood).
> >
> > Please see the discussion above titled “Why is style robust to DPO optimization?”. To summarize:
> >  The representations used by StyleDetect extract features discriminative of individual low-resource authors (those with 100 posts or less), which LLMs struggle to replicate, unlike common styles or high-resource authors (e.g., Shakespeare).
> > Optimizing against a generic "machine" style is under-defined; without a specific human author or concrete style as a target, it's impossible to choose a single successful optimization goal because human writing spans such a wide range.
> >
> > > The proposed humaniser (as described in section 3.1) is an interesting, simple idea. Could you put it in more context as to how it relates to previous paraphrasers. Has a similar idea been tried (successfully or otherwise) elsewhere? Is it generally seen as improving the quality of machine generated text to human eyes? I think this is the strongest idea in the paper, I would like to use the fact that 1) the humanizer is novel and 2) your humaniser could find applications outside of detecting machine generated text as points in your favour, but I'd like you to provide a little more evidence/context for me to do so.
> >
> > Please see the discussion above titled “Novelty of our style-aware paraphraser”. To summarize, our style-aware paraphraser increases the diversity of the machine-generated text, is the first of its kind to be applied to re-writing machine-generated text, and outperforms the previous state-of-the-art (TinyStyler) in the task of style transfer. Due to its good performance in style-transfer, our paraphraser might find further applications in increasing the diversity of synthetic datasets and it could be applied to more classical tasks such as anonymization.
> >
> >
> > > After I wrote the previous paragraph I searched for other paraphrase inverters. I found https://aclanthology.org/2025.findings-acl.227/ . Mitigating Paraphrase Attacks on Machine-Text Detection via Paraphrase Inversion Rafael Alberto Rivera Soto, Barry Y. Chen, Nicholas Andrews. I think this should be cited, and I think it's the same essential idea for evading detection, could you comment.
> >
> > Note that this paper was cited in line 267. That work focuses on inverting a paraphraser’s output to improve detection, whereas in our work we focus on designing a better, more diverse paraphrase capable of mimicking many human styles.
> >
> > > I don't think your claim in the 'main contributions' section that 'optimizing against such detectors does not reduce their performance' is accurate. What you are really saying is that the strategy for optimizing against log-likelihood based detectors does not translate directly to style-based detectors.
> >
> > We  agree with the reviewer’s assessment, and have modified our manuscript to reflect this change. As mentioned above, we consider more than only log-likelihood based detectors and we do evaluate against supervised detectors (RADAR and ReMoDetect) as well. However, there might be other zero-shot detectors that don’t just look at averages of log-likelihood, and therefore we change the language in our revised manuscript to “popular zero-shot and supervised detectors”.
> >
> > > 'These results suggest that the features indicative of writing style are distinct from those used by detectors that use features derived from the predicted conditional distributions.' I think this claim is slightly too strong. I would replace 'features derived from the predicted conditional distributions' with 'log-likelihood'.
> >
> > We agree with the reviewer’s assessment that other more clever zero-shot detectors might indeed look at features that overlap with those of stylistic detectors. As such, we’ve revised our manuscript to reflect this change, specifying “popular zero-shot and supervised detectors”.

---

> > > ### Author Response · Authors · 2025-11-21
> > >
> > > > I would like to know how long the typical texts you are dealing with are (this is useful info when looking at the effectiveness of detectors)
> > >
> > > We provide the number of tokens below. Note that we ensure that all the human texts are between 32-128 tokens long according to the FacebookAI/roberta-base tokenizer. Moreover, when we generate machine-text, we specify: “Keep the response around <LENWORDS> words”, setting <LENWORDS> to the number of words of the original human text. We found this to keep the length distributions between the machine-generated and human-written text similar. See Appendices F and H.3 for more details on the prompts.
> > >
> > >
> > >
> > > | **Dataset** | **Number of Tokens**      |
> > > |-------------|----------------------------|
> > > | Reddit      | 57.89 (29.53)              |
> > > | Amazon      | 74.73 (39.97)              |
> > > | Blogs       | 103.64 (34.85)             |
> > >
> > > **Table:** Average number of tokens using the `Mistral-7B-Instruct` tokenizer (standard deviation in parentheses) for the machine-generated text.
> > >
> > >
> > > > What decoding strategies (temp, top-p, top-k etc.) are you using when you generate text from machine. Note, until recently huggingface enabled top-k=50 by default so unless this has been explicitly controlled for, you may be using top-k generation.
> > >
> > > We use a top-p of 0.9 and a temperature value of 0.7. We’ve added these details to the manuscript.
> > >
> > > > You write 'We find that our approach preserves the meaning of the original text (similarity >0.85)'. Is this a fair conclusion? I have no idea how to interpret it, but note that it is worse than some of the alternative methods.
> > >
> > > We note that our approach achieves a higher semantic similarity than DIPPER, another paraphraser widely used and accepted by the community (Table 3 of the manuscript and reproduced below). Furthermore, we note that it has been found that semantic similarity as measured by SBERT does correlate features usually thought of as “stylistic”[2]. As our paraphraser re-writes text in differing styles, part of the measured performance might come from modifying the style.
> > >
> > > | **Methods →**     | Prompting        | Paraphrasing     | DIPPER           | TinyStyler       | Ours            |
> > > |-------------------|------------------|------------------|------------------|------------------|-----------------|
> > > | **Edit Distance** | 134.05 (81.52)   | 156.57 (74.50)   | 227.39 (117.94)  | 212.58 (101.71)  | 199.09 (94.25)  |
> > > | **Semantic Sim.** | 0.91 (0.11)      | 0.93 (0.07)      | 0.84 (0.11)      | 0.78 (0.13)      | 0.85 (0.12)
> > >
> > >
> > >
> > > > I found it really hard to understand what is meant by 'sample size' in your article. I think the answer is that, when sample size is n, you compute for example the fast detect score of n separate human texts and then compute the mean fast detect score of those n human texts, compared to the mean fast detect score of n machine texts. This is quite an unusual setup, it's not necessarily bad because it's more or less equivalent to just increasing the number of tokens to help detectors, but because we're so primed to imagine you submitting one text for analysis I think you need to go to more effort to stress exactly what you're doing.
> > >
> > > The reviewer’s understanding is correct. Note that we discuss this setup in Section 6.1 of the manuscript.
> > >
> > > > I think the caption to figure 2 is confusing. When you say detection performance (lower is better), you mean that higher is better for the detector, but lower is better for your paraphraser. I would write something like 'a lower score indicates better performance of the rewriter'.
> > >
> > > We agree that the caption could’ve been clearer and have made the appropriate modifications in the revised manuscript.

---

> ### Author Response · Authors · 2025-11-21
>
> > Could you mention text lengths. There is a known problem with the way that Fast-Detect normalises by passage length which can artificially inflate or depress it's performance, see for example TempTest: Local Normalization Distortion and the Detection of Machine-generated Text section 8.2.
>
> We provide the number of tokens below. Note that we ensure that all the human texts are between 32-128 tokens long according to the roberta-large tokenizer. Moreover, when we generate machine-text, we specify: “Keep the response around <LENWORDS> words”, setting <LENWORDS> to the number of words of the original human text. We found this to keep the length distributions between the machine-generated and human-written text similar. See Appendices F and H.3 for more details on the prompts.
>
> | **Dataset** | **Number of Tokens**      |
> |-------------|----------------------------|
> | Reddit      | 57.89 (29.53)              |
> | Amazon      | 74.73 (39.97)              |
> | Blogs       | 103.64 (34.85)             |
>
> **Table:** Average number of tokens using the `Mistral-7B-Instruct` tokenizer (standard deviation in parentheses) for the machine-generated text.
>
>
> [1] Rivera Soto, et. al Learning Universal Authorship Representations. EMNLP 2021
>
> [2] Andrew Wang, Cristina Aggazzotti, Rebecca Kotula, Rafael Rivera Soto, Marcus Bishop, Nicholas Andrews; Can Authorship Representation Learning Capture Stylistic Features?. Transactions of the Association for Computational Linguistics 2023; 11 1416–1431

---

> > ### Comment · Reviewer_ZPpT · 2025-11-24
> > **Thanks for Responses**
> >
> > Many thanks for your responses, I think I now have a clearer view of the manuscript. I will read it again completely before going in to the discussion phase with other reviewers (which I enter with an open mind).

---

### Official Review · Reviewer_fRsS · 2025-10-31

**Soundness:** 2
**Presentation:** 2
**Contribution:** 2
**Rating:** 2
**Confidence:** 4

**Summary:**

Authors of this paper explore the importance of text style in the detection of AI-generated texts and study the effect of the attacks on detectors via direct optimization of the generating model.
They propose a method of paraphrasing machine-generated texts that ``enforces'' the style of a human author on them to avoid detection; they further strengthen it with direct preference optimization against a detector. Authors conduct a series of experiments on data from 3 domains against various detectors and compare the proposed approach against competitive solutions. Finally they perform analysis of the learned writing style representations.

**Strengths:**

- The results of the experiments conducted by the authors demonstrate that their approach outperforms other paraphrasing attacks at avoiding automatic detection.

- The experiments cover rather underexplored setup of detection AI-generated content by analyzing multiple text samples from the same author.

- Limitations of the proposed method are well outlined.

**Weaknesses:**

- The fact that existing methods for AI-generated texts are very brittle to paraphrasing is quite well-known; this paper does not provide any novelty in this regard. Essentially, the main contribution lies in applying existing methods to transfer the style of a human author to machine generated texts.

- Main claims of the paper contradict each other: "although LLMs can be optimized to defeat machine-text detectors, they remain identifiable by detectors that avail of writing style and that moreover, optimizing against such detectors does not reduce their performance", and, at the same time, results of the experiments in Section 5 clearly show that optimizing against writing style-avail detectors via the proposed method clearly reduces their performance.

- ``Since StyleDetect requires exemplars from the machine class, we provide 100 examples from the unoptimized LLM model`` (section 2, lines 139-140). By this procedure, StyleDetect is initialized with the information on the generating model and the text domain, while other methods (FastDetectGPT, Binoculars) are not; this makes the comparison somewhat unfair and makes the results in Table 1 less informative.

- Only a limited range of data domains was covered by the evaluation (free-form texts from Amazon/Reddit/...). More diverse domains, like Wikipedia articles, poetry, news, or some Q&A would help better analyze the generability of the proposed method.

- Certain important implementation details are omitted or moved to the appendices; it is somewhat difficult to grasp the exact pipeline of the proposed method. In particular, what is the detector used in the direct preference optimization of the proposed method (line 185)?

**Questions:**

Questions/Suggestions:

- Please, clarify "N" in the axis labels on the plots.

- In Tables 3 and 4 could you please provide standard deviations for the average semantic similarities?

- Binoculars is a general method that leverages representations from a pair of related models to detect AI-generated texts. Which model pair/pairs were used in the experiments?

---

> ### Author Response · Authors · 2025-11-21
>
> We thank the reviewers for their thoughtful comments and constructive feedback. Note that we’ve updated our manuscript, the section numbers referred to in the rebuttal are those of the revised manuscript. To facilitate the discussion, we briefly summarize our three primary contributions. First, we provide a comprehensive analysis demonstrating that current DPO-optimized attacks (e.g., Nicks et al.) fail to remove stylistic "fingerprints," rendering them vulnerable to style-based detection. Second, we demonstrate that even when DPO is used to optimize specifically against style-based detectors, the outputs still remain identifiable by said detectors. Third, we introduce a novel Style-Aware Paraphraser that bridges the gap between style transfer and evasion by simultaneously optimizing for specific authorship styles and undetectability, achieving state-of-the-art performance when compared against other attacks (Figure 2, Section 6.1) and in the traditional task of style-transfer (Section 6.3).
>
> Weaknesses:
>
> > The fact that existing methods for AI-generated texts are very brittle to paraphrasing is quite well-known; this paper does not provide any novelty in this regard. Essentially, the main contribution lies in applying existing methods to transfer the style of a human author to machine generated texts.
>
> We wish to clarify that our contribution goes beyond applying existing style transfer methods. As our experiments demonstrate, existing style transfer methods (like TinyStyler) fail to evade detection (Figure 2), and existing evasion methods (like Nicks et al.) fail to change the underlying style (Section 3). Our novelty lies in bridging this gap. We introduce a training recipe that simultaneously optimizes for specific authorship style and undetectability via DPO. This joint optimization is non-trivial, has not been explored in prior work, and is necessary to defeat the new class of style-based detectors. Moreover, our style-aware paraphraser outperforms the previous state-of-the-art in the traditional task of style-transfer (Section 6.3)
>
> > Main claims of the paper contradict each other: "although LLMs can be optimized to defeat machine-text detectors, they remain identifiable by detectors that avail of writing style and that moreover, optimizing against such detectors does not reduce their performance", and, at the same time, results of the experiments in Section 5 clearly show that optimizing against writing style-avail detectors via the proposed method clearly reduces their performance.
>
> We clarify that the perceived contradiction is actually the central motivation of our study.
> Claim 1 (analysis of Nicks et al.): Note that one of our main claims is that the DPO-optimized LLMs proposed by Nicks et al. (ICLR 2024) are still eminently detectable by detectors that avail of writing style (shown in Section 3).
> Claim 2 (our method): Because of claim one, we developed a new method. Our  style-aware paraphraser re-writes text in the style of human authors while avoiding the telltale signs used by common detectors. Our style-aware paraphraser successfully succeeds in altering the stylistic signature, which is why it successfully degrades the performance of style-based detectors in Section 5.
>
> Finally, the title of our paper implies that we show that existing optimization methods don’t change the style, and that we show how to change it: “Language Models Optimized to Fool Detectors Still Have a Distinct Style (*And How to Change It*)”.
>
> > Since StyleDetect requires exemplars from the machine class, we provide 100 examples from the unoptimized LLM model (section 2, lines 139-140). By this procedure, StyleDetect is initialized with the information on the generating model and the text domain, while other methods (FastDetectGPT, Binoculars) are not; this makes the comparison somewhat unfair and makes the results in Table 1 less informative.
>
> StyleDetect is a few-shot method that requires examples from the machine class. Some of the detectors we evaluate are zero-shot (Binoculars, FastDetectGPT, Rank, LogRank), others avail of supervised training (RADAR, ReMoDetect), and others avail of a few examples of the machine class (StyleDetect). The results in Figure 2 evaluate across all detectors, showing that our approach best evades detection even under the assumption that samples from the unoptimized model are available.
>
> Moreover, as the reviewer stated, StyleDetect is initialized with samples from the unoptimized model. If the optimization procedure would’ve been effective, then the distribution should’ve shifted so as to make StyleDetect ineffective. The fact that it doesn’t further reinforces our point that DPO doesn’t change the style of the original unoptimized model.

---

> > ### Author Response · Authors · 2025-11-21
> >
> > > Only a limited range of data domains was covered by the evaluation (free-form texts from Amazon/Reddit/...). More diverse domains, like Wikipedia articles, poetry, news, or some Q&A would help better analyze the generability of the proposed method.
> >
> > We selected Reddit, Amazon, and Blogs because they represent high-entropy, distinct authorship styles, which constitutes the primary threat model for impersonation and style-based evasion. Domains like Wikipedia and news enforce a uniform "encyclopedic" style that suppresses individual authorship features, making them less relevant for evaluating style transfer.
> > However, we highlight that the Reddit domain is stylistically vast. It ranges from highly informal, slang-heavy text (e.g., r/WallStreetBets) to formal, structured political discourse (e.g., r/Europe), providing a rigorous test of the model's generalization capabilities.
> >
> >
> > > Certain important implementation details are omitted or moved to the appendices; it is somewhat difficult to grasp the exact pipeline of the proposed method. In particular, what is the detector used in the direct preference optimization of the proposed method (line 185)?
> >
> > We have added the missing details to the main text. Specifically, the detector used for DPO in our proposed method is a RoBERTa-base classifier trained to distinguish between our system's outputs and human text.
> >
> > Questions:
> >
> > > Please, clarify "N" in the axis labels on the plots.
> >
> > We have updated the axis labels to "Number of Samples" for clarity.
> >
> > > In Tables 3 and 4 could you please provide standard deviations for the average semantic similarities?
> >
> > We have added standard deviations (in parenthesis) to Tables 3 and 4 in the manuscript. For convenience, we reproduce the data here:
> >
> > | **Methods →**     | Prompting        | Paraphrasing     | DIPPER           | TinyStyler       | Ours            |
> > |-------------------|------------------|------------------|------------------|------------------|-----------------|
> > | **Edit Distance** | 134.05 (81.52)   | 156.57 (74.50)   | 227.39 (117.94)  | 212.58 (101.71)  | 199.09 (94.25)  |
> > | **Semantic Sim.** | 0.91 (0.11)      | 0.93 (0.07)      | 0.84 (0.11)      | 0.78 (0.13)      | 0.85 (0.12)
> >
> >
> > > Binoculars is a general method that leverages representations from a pair of related models to detect AI-generated texts. Which model pair/pairs were used in the experiments?
> >
> > We use the same pair of LLMs as explored in the original Binoculars paper, namely tiiuae/falcon-7b and tiiuae/falcon-7b-instruct.

---

### Official Review · Reviewer_naRM · 2025-10-31

**Soundness:** 3
**Presentation:** 3
**Contribution:** 3
**Rating:** 8
**Confidence:** 4

**Summary:**

The paper investigates whether language models (LLMs) can evade AI-text detectors and whether such evasion eliminates all detectable signals. When LLMs are explicitly optimized against detectors (e.g., FastDetectGPT), their outputs retain identifiable stylistic patterns that distinguish them from human writing. Authors later introduce a style-aware paraphraser, trained via supervised fine-tuning and DPO, which mimics human writing styles to avoid being detected. The method improves over prior paraphrasing and prompting attacks, producing text that is nearly indistinguishable from human-written samples when judged individually, though detectability re-emerges when multiple samples from the same source are aggregated. The study concludes that style-based features remain robust against current optimization attacks, and reliable AI-text detection may only be feasible when based on multiple documents rather than single samples.

**Strengths:**

- The paper provides strong empirical evidence that style-based detection methods remain robust even after preference tuning aimed at fooling detectors.
 - The style transfer experiments are valuable and demonstrate practical potential for improving the naturalness and “humanization” of LLM-generated text, particularly for applications like chatbots.
 - Overall, the work offers a well-executed empirical study that contributes both to improving the human-likeness of generated text and to understanding how to detect it more reliably.

**Weaknesses:**

The weaknesses are minor, mostly text-based.
- There could be a more in-depth intro to the style-based detectors, for example, explaining what is the style feature space.
- The paper could benefit from studying larger LLMs, say up to 32B, without fine-tuning, to give a perspective on the limits of the style-based detection applicability.

**Questions:**

- Do these results generalize to larger LLMs, say GPT-4o?
- Do you perform any human evaluation to confirm that reduced detectability corresponds to genuinely more human-like writing, rather than merely adversarial surface changes?

---

> ### Author Response · Authors · 2025-11-21
>
> We thank the reviewers for their thoughtful comments and constructive feedback. Note that we’ve updated our manuscript, the section numbers referred to in the rebuttal are those of the revised manuscript. To facilitate the discussion, we briefly summarize our three primary contributions. First, we provide a comprehensive analysis demonstrating that current DPO-optimized attacks (e.g., Nicks et al.) fail to remove stylistic "fingerprints," rendering them vulnerable to style-based detection. Second, we demonstrate that even when DPO is used to optimize specifically against style-based detectors, the outputs still remain identifiable by said detectors. Third, we introduce a novel Style-Aware Paraphraser that bridges the gap between style transfer and evasion by simultaneously optimizing for specific authorship styles and undetectability, achieving state-of-the-art performance when compared against other attacks (Figure 2, Section 6.1) and in the traditional task of style-transfer (Section 6.3).
>
> Weaknesses:
>
> > There could be a more in-depth intro to the style-based detectors, for example, explaining what is the style feature space.
>
> We agree that this would be a great addition to our paper. With the additional page allowed for the response, we’ve added a new section titled “Preliminaries: Style Representations” to our paper that introduces the notion of the stylistic feature space.
>
> > The paper could benefit from studying larger LLMs, say up to 32B, without fine-tuning, to give a perspective on the limits of the style-based detection applicability.
>
> > Do these results generalize to larger LLMs, say GPT-4o?
>
> We appreciate the suggestion to extend our evaluation to larger LLMs. We believe our method would generalize well to these models and will strive to demonstrate this during the rebuttal. If time constraints prevent a full analysis now, we commit to including these experiments in the camera-ready version.
>
> > Do you perform any human evaluation to confirm that reduced detectability corresponds to genuinely more human-like writing, rather than merely adversarial surface changes?
>
> We do not perform direct human evaluations; however, we note that the StyleDistance[1] representations (used in Sections 6.1 and 6.2) have been previously validated by humans. Thus, the ability to close the gap in this representation space serves as evidence that our approach generates text that is more human-like, specifically in terms of the ability of these stylistic representations to characterize variance in human writing style.
>
> [1] Patel, et. al. STYLEDISTANCE: Stronger Content-Independent Style Embeddings
> with Synthetic Parallel Examples. ACL 2025

---

### Meta-Review · Area_Chair_x5md · 2025-12-22

**Summary:**

The major concerns focus on the significance and novelty of the work, clarity of the method and experimental setup, generalizability to larger LLMs and broader domains, and unclear evaluation reporting.

**Reviewer Concerns:**

Some concerns were addressed in the rebuttal, such as those related to clarity of the claims and implementation details. However, major concerns remain outstanding, such as significance and novelty of the work.

**Reviewer Scores:**

Reviewer naRM is likely to keep the positive score. Reviewer fRsS may increase the score slightly. However, Reviewer ZPpT and 3Y7h may not increase the scores as the major concerns were about the significance and novelty of the work.

---

### Decision · Program_Chairs · 2026-01-26

Reject